# Molecular characterization of cell types in the squid *Loligo vulgaris*

**Jules Duruz[1], Marta Sprecher[1], Jenifer C Kaldun[1], Al-Sayed Al-Soudy[1], Heidi EL Lischer[2], Geert van Geest[2], Pamela Nicholson[3], Rémy Bruggmann[2], Simon G Sprecher[1]\***

[1]Department of Biology, Institute of Zoology, University of Fribourg, Fribourg, Switzerland; [2]Interfaculty Bioinformatics Unit and Swiss Institute of Bioinformatics, University of Bern, Bern, Switzerland; [3]Institute of Genetics, University of Bern, Bern, Switzerland

**Abstract** Cephalopods are set apart from other mollusks by their advanced behavioral abilities and the complexity of their nervous systems. Because of the great evolutionary distance that separates vertebrates from cephalopods, it is evident that higher cognitive features have evolved separately in these clades despite the similarities that they share. Alongside their complex behavioral abilities, cephalopods have evolved specialized cells and tissues, such as the chromatophores for camouflage or suckers to grasp prey. Despite significant progress in genome and transcriptome sequencing, the molecular identities of cell types in cephalopods remain largely unknown. We here combine single-cell transcriptomics with in situ gene expression analysis to uncover cell type diversity in the European squid *Loligo vulgaris*. We describe cell types that are conserved with other phyla such as neurons, muscles, or connective tissues but also cephalopod-specific cells, such as chromatophores or sucker cells. Moreover, we investigate major components of the squid nervous system including progenitor and developing cells, differentiated cells of the brain and optic lobes, as well as sensory systems of the head. Our study provides a molecular assessment for conserved and novel cell types in cephalopods and a framework for mapping the nervous system of *L. vulgaris*.

## Editor's evaluation

This article describes cell types in the head of the squid, *Loligo vulgaris*, through expression patterns of key genes identified in single-cell transcriptomics. This topic is generally of great comparative interest. The data presented here are convincing, and these valuable findings will contribute to a better understanding of the cephalopod nervous and sensory systems, providing a basis for future comparative and evolutionary research.

## Introduction

Cephalopods are part of a long tradition in evolutionary neuroscience and are often highlighted as an ideal example of convergent evolution of nervous systems (*Young, 1965*; *Young, 1985*; *Abbott et al., 1995*). Their large brains, relative to body size, and behavioral repertoires offer opportunities for comparative analysis of the evolution of learning and memory mechanisms. Cephalopods possess a sophisticated nervous system that resembles that of vertebrates in relative size and cognitive abilities (*Young, 1991*; *Hochner et al., 2006*). These complex behavioral abilities include various forms of learning, problem-solving, and body color and pattern changes used either as predatory or anti-predatory behaviors or for social communication (*Fiorito et al., 1990*; *Langridge et al., 2007*; *Staudinger et al., 2013*; *Darmaillacq et al., 2014*; *Schnell et al., 2015*; *Richter et al., 2016*; *Lin and*

*For correspondence:
simon.sprecher@unifr.ch

**Competing interest:** The authors declare that no competing interests exist.

*Chiao, 2017*; *Marini et al., 2017*; *Mather and Dickel, 2017*; *Hanlon and Messenger, 2018*). The fossil record suggests that cephalopods were more prominent in marine ecosystems over 300 million years ago with a higher biodiversity than today. It is assumed that the rise of bony fishes and resulting competition for resources in that ecological niche played a major role in the extinction of many clades of cephalopods, including all the shelled cephalopods except few members of the subclass Nautiloidea. Today, the class Cephalopoda has over 800 current living species with extremely diversified lifestyles and body shapes (*Kröger et al., 2011*). The vast majority of current cephalopod species belong to the soft-bodied subclass Coleoidea (cuttlefishes, octopuses, and squids), and a small handful belong to the subclass Nautiloidea (nautiluses). Coleoid cephalopods, which have internalized or lost their shell over evolutionary time, were able to swim more easily and could compete with other predators in open waters. The continuous competition with other animals to access resources has been proposed to have been the main trigger for the selection of cephalopods that could develop specialized strategies for accessing resources unavailable to competing species. In coleoid cephalopods, this evolutionary process resulted in larger brains, camera-type eyes, as well as abilities to voluntarily change their skin pattern and texture for camouflage and social communication (*Young, 1973*; *Hanlon and Messenger, 2018*; *Amodio et al., 2019*).

The European squid *Loligo vulgaris* is well known for its economic role in the food and fishing industry but is much less studied for its fascinating biology. This species is known for its striking ability for camouflage typically in open-water environments by the use of its specialized iridescent cells to reflect light in different manners depending on its surroundings. The neuroanatomy and nervous system morphology of *L. vulgaris* have been described in detail in histological studies (*Young, 1976*; *Wild et al., 2015*). Detailed staging of the embryonic development of the related species *Loligo pealii* was established and can be used as a reference for *L. vulgaris* (*Arnold, 1965*). While the gross anatomy of squids is well known from histology and morphological studies, we are currently lacking resolution at the molecular level to identify genes that specify the many cell types that make up the unique cephalopod body-plan and nervous system. Since the publication of the first complete genome sequence of *Octopus bimaculoides*, there has been increasing interest in cephalopod biology and neuroscience. In addition, the availability of genomic sequences of other cephalopods, such as *Euprymna scolopes* (Sepiida), *Architeuthis dux* (Oegopsida), and *Nautilus pompilius* (Nautilida), opens the possibility for molecular comparisons between these distant clades of cephalopods with drastically different morphologies and lifestyles. These comparisons constitute a major step for understanding the molecular basis of cephalopod evolution (*Albertin et al., 2015*; *Belcaid et al., 2019*; *da Fonseca et al., 2020*; *Zhang et al., 2021*).

Single-cell transcriptomics experiments in different cephalopods have shed light on the cell type diversity of the *O. bimaculoides* visual system, the brain of *Octopus vulgaris,* and the nervous system of the squid *Euprymna berryi* (*Gavriouchkina et al., 2022*; *Songco-Casey et al., 2022*; *Styfhals et al., 2022*).

To gain further insights into the cellular diversity and cell type-specific gene expression in squids, we performed single-cell transcriptomics on whole *L. vulgaris* heads (stages 28–30 based on *Arnold, 1965*, pre-hatchling stage). Additionally, we used hybridization chain reaction in situ hybridization (HCR ISH) to visualize gene expression in situ and correlate single-cell transcriptomes with anatomical features. By combining these methods, we were able to identify and assess common cell types found in most bilaterians, such as different types of neurons, muscles, and connective tissues. Interestingly, even in these canonical tissues, we found certain clusters that highly express cephalopod-specific genes, as exemplified by some subtypes of connective tissues. Thus, while cells can be assigned a certain identity based on well-characterized marker genes, such as collagens, these cell types have likely further diverged in cephalopods. Neural tissues are prominent in the head and include the brain, optic ganglia, eyes, and other sensory organs. We identified and analyzed new markers of different cells in the nervous system, including widely expressed pan-neuronal genes, genes marking distinct stages of neural differentiation, markers for neural subsets, as well as sensory cell types. In addition, we also characterized cephalopod-specific cell types, such as the chromatophores or cells of the suckers. Our study provides new insights into the transcriptomic signatures of cell types in *L. vulgaris*. By analyzing the expression patterns of many cell type-specific genes, we assessed the conservation of features previously described in other animal clades but also previously unknown features and modalities of cell types that appear specific for cephalopods.

## Results

### Single-cell transcriptome of the *L. vulgaris* head

To generate a reference transcriptome of *L. vulgaris,* embryos were collected at late embryonic stages for RNA extraction and subsequent cDNA synthesis. The transcriptome was sequenced, assembled de novo, and annotated (for details, see 'Materials and methods').

Single-cell RNA sequencing was performed from the dissociated cells of the whole heads of *L. vulgaris* pre-hatchlings. Cells were captured and sequenced in two different experiments and the resulting barcoded sequences were mapped to the reference transcriptome. The mapping rate of the sequences was of 29.3% and 29.8% for samples 1 and 2, respectively (*Figure 1—figure supplement 1A*). This low mapping rate can be explained by *L. vulgaris* genes not being mapped to the transcriptome because the genes are missing from the transcriptome and by the presence of cells and ambient RNA from other species in the single-cell suspension. The assessment of the transcriptome assembly using benchmark universal single-copy orthologs (BUSCO) showed a score of 54.4% (49.8% single-copy, 4.6% duplicated, 1.9% fragmented), which indicates that a number of genes are likely missing from our transcriptome that in turn may impact downstream analysis. Cells that were identified and sequenced were clustered and analyzed. 20,000 cells were analyzed in our study. Cells were filtered to keep only the ones with a gene-per-cell count comprised between 200 and 2000 (median = 700 genes/cell). This filtering step resulted in a total of 19,974 cells. Cells were plotted to identify the presence of outliers for gene-per-cell count, UMI-per-cell count, or percentage of mitochondrial genes (*Figure 1—figure supplement 1B*). Clustering was performed with Seurat using 25 principal components with a resolution value of 2. This resulted in a total of 34 cell clusters numbered from 0 to 33. Clusters were assigned to presumed cell type identities based on the genes differentially expressed in each cluster (Cl). Clusters were manually annotated and fitted into cell type categories based on the expression of specific markers (*Figure 1B*, *Figure 1—figure supplement 2*). These defined cell type categories include muscle, connective tissue, stem cells, epidermis, papilin[+] cells, neurons, foxD1[+] cells, reflectin[+] cells, ASc[+] cells, cilia-related cells, photoreceptors, sucker, and lens.

The identification of muscle cells is based on the expression of various genes involved in the formation of contractile fibers such as *myosin heavy chain, troponin, myosin regulatory light chain, paramyosin, tropomysin* (Cl00). A second muscle cluster (Cl25) expresses markers including *myosin catalytic light chain* and *paramyosin* but is also distinguished by the expression of the transcription factor *soxE*. Smooth muscle-like cells (Cl25) are characterized by the expression of the homeobox transcription factor *nkx2-5,* which plays a crucial role in cardiac development in human embryos and *angiopoietin-1 receptor,* involved in angiogenesis in many species (*Elliott et al., 2006*). Because of the presence of these two markers, we hypothesize that these cells could be a part of the squid circulatory system.

Connective tissues are defined by the very broad expression of many different types of collagens, which are one of the main components of connective tissues (Cl01, Cl06, Cl23, Cl28). The presence of enriched *laminin* and *fibroblast growth factor receptor* further suggests that these cell clusters include precursors or differentiated cells of connective tissues. These markers that are typical of connective tissues are, however, co-expressed with several uncharacterized genes that have known orthologs in other cephalopod species. These previously unidentified markers could provide new avenues for the study of cephalopod connective tissues and provide a molecular complement to the existing morphological data (*Thompson and Kier, 2001*; *Kier and Stella, 2007*).

Cells of the epidermis are characterized based on the expression of an *egf-like* epidermal growth factor (Cl08, Cl09). Interestingly, Cl08 comprises chromatophores defined by the co-expression of *tyrosinase* and *tryptophan 2,3-dioxygenase*, two members of the melanin synthesis pathway that, when knocked out, have been shown to result in depigmentation in *Doryteuthis pealeii* (*Crawford et al., 2020*). Cl31 was characterized by a different epithelial marker *epithelial splicing regulatory protein* (*esrp*).

Three clusters (Cl03, Cl04, Cl12) highly express *papilin,* a gene which was suggested to play a role in the formation of extracellular matrix in *Drosophila melanogaster* (*Fessler et al., 2004*; *Apte, 2020*). The function of this gene in other clades is, however not known.

Cells of the suckers (Cl29) could be identified by the enriched expression of *suckerin* genes that have been shown to be expressed in the sucker ring teeth of squids (*Kumar et al., 2016*).

Neurons were identified based on widely used markers, including two genes belonging to the ELAV RNA-binding protein family *elav-like 1* and *elav-like 2* (Cl05, Cl07, Cl11, Cl13, Cl15). Known

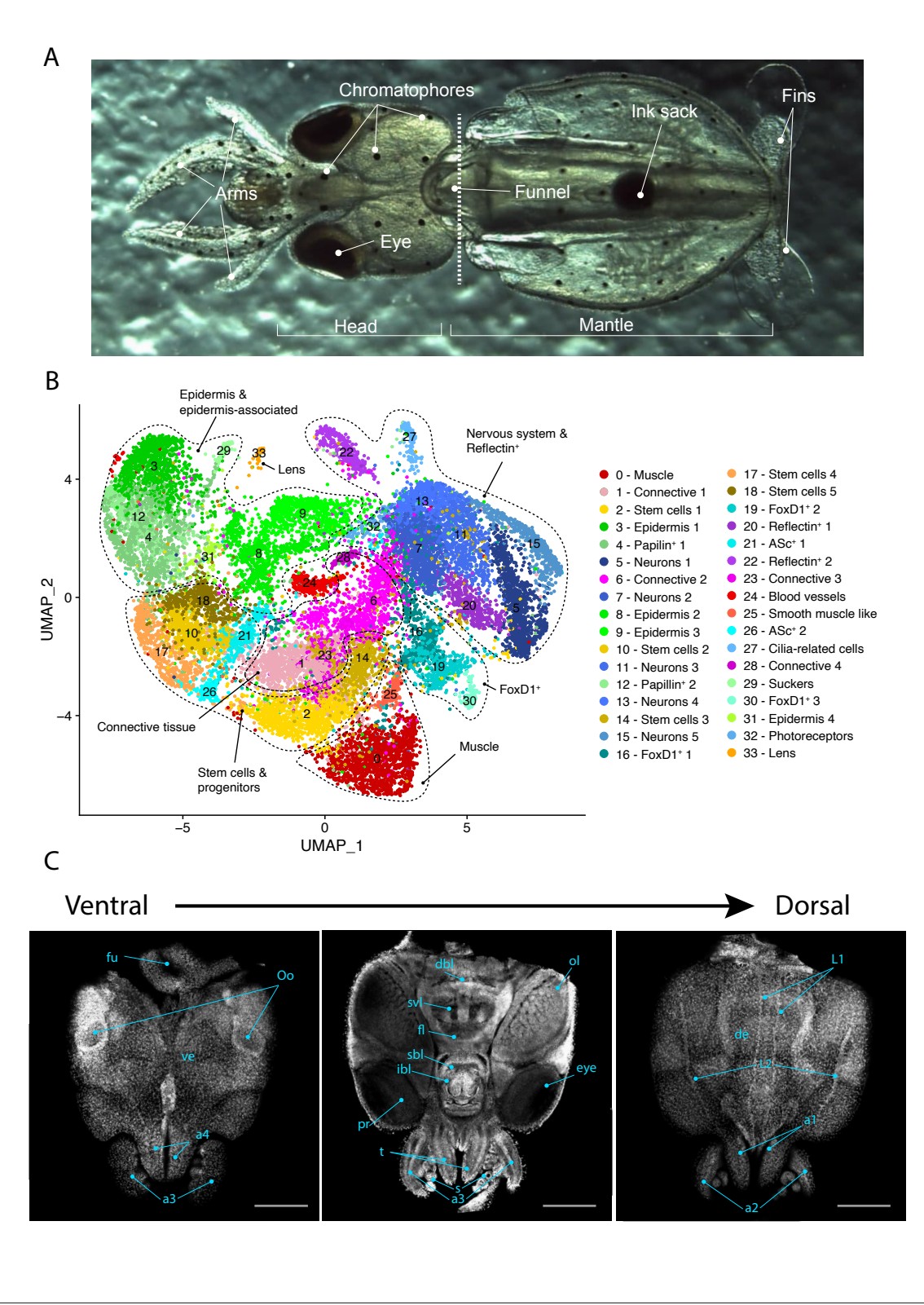

**Figure 1.** Single-cell transcriptome of the *L. vulgaris* head. (**A**) Ventral view of a recently hatched *L. vulgaris* with relevant external anatomical structures labeled. The dotted line represents the plane of amputation. The region anterior to this line is what was used in this study and is referred to as the head region. (**B**) Two-dimensional UMAP representing the 34 cell clusters and their cell type characterization based on marker gene expression. These defined cell type categories include muscle, connective tissue, stem cells, epidermis, papilin+ cells, neurons, foxD1+ cells, reflectin+ cells, ASc+ cells, cilia-related

*Figure 1 continued on next page*

*Figure 1 continued*

cells, photoreceptors, sucker, and lens. (**C**) Successive confocal micrographs of the ventral–dorsal view of *L. vulgaris* hatchling heads stained with DAPI (nuclei). The entire ventral–dorsal surface of the head of *L. vulgaris* is labeled with relevant anatomical structures: (a) labels a1–4 = arms 1–4, (bl) brachial lobe, (dbl) dorsal basal lobe, (de) dorsal epidermis, (fl) frontal lobe, (fu) funnel, (ibl) inferior buccal lobe, (L1 ,L2) epidermal lines, (Oo) olfactory organ, (ol) optic lobe, (pr) proximal segment of retinas, (s) suckers, (sbl) superior buccal lobe, (svl) subvertical lobe, (t) tentacles, and (ve) ventral epidermis. Scale bars = 500 μM.

The online version of this article includes the following figure supplement(s) for figure 1:

**Figure supplement 1.** Quality controls.

**Figure supplement 2.** Marker genes.

markers for the presynaptic machinery such as *synaptobrevin, synaptogyrin, synaptophysin,* and *synapse-associated protein 1* are also broadly expressed in these clusters as well as *amyloid beta-binding-like, tetraspanin-8,* and *rab3*.

Cl27 is characterized by the expression of different cilia-related and flagella-related markers, together with the neuronal gene *rab3* and *synaptogyrin-1*. The modality of these cells remains unknown, but the presence of cilia/flagella-related genes together with neuronal markers could suggest a mechanosensory function (see Figure 4).

Photoreceptors (Cl32) could be identified based on the expression of various members of the canonical phototransduction cascade, including *trp-like, visual arrestin, retinal binding protein, phospholipase C,* and *rhodopsin*.

Cl16, Cl19, and Cl30 are very clearly defined by the expression of the forkhead box transcription factor *foxD1*. In other phyla, this gene is involved in various developmental processes. However, because the presence of this gene is not sufficient to affirm a specific cell type, we named these cells foxD[+] cells. A large proportion of these cells co-express the transcription factor *ets4*. In Cl30, *foxD1* was co-expressed with a *GATA zinc-finger protein (gata-2)* and a few blood-related markers (*ferritin, hemocyte, apolipophorin*).

Cl20 and Cl22 are characterized by the expression of different types of *reflectin* genes, a family of proteins used in cephalopods to control light reflection by modulating the iridescence of the skin (*Crookes et al., 2004*). Interestingly, the transcriptional repressor *scratch-2,* which is highly expressed in several neuronal clusters, is also highly expressed in reflectin[+] cells. We also observe expression of reflectins in neurons in agreement with this notion.

Different types of neural progenitors (Cl21, Cl26) are characterized by the high expression of a *soxB1* and an *achaete-scute* (*asc*). Genes of the *achaete-scute* complex have been broadly studied for their implication in the specification of the neuronal lineage, particularly in the fruit fly (*Campuzano and Modolell, 1992*; *Skeath and Doe, 1996*; *Zhao et al., 2007*; *Arefin et al., 2019*; *Soares et al., 2022*). Both clusters also express histone-related genes and markers of proliferation, suggesting that they may be replicating undifferentiated cells. The presence of a *soxB1* in these cells indicates that they may be proliferating neuronal progenitors, based on previous evidence of expression of soxB1 genes during neurogenesis in cephalopods (*Miyagi et al., 2009*; *Focareta and Cole, 2016*).

The lens is a specialized part of the eye for which we identified a cluster (Cl33) based on the expression of *s-crystallin,* an important structural component of the tissue of the lens (*Tomarev and Piatigorsky, 1996*).

To next move beyond the in silico assessment of cell cluster identity and describe where distinct cell types are localized, we analyze the expression of an array of marker genes specifically enriched in cell clusters of particular interest using HCR ISH. We further assess the newly identified cell types and describe their localization anatomically (*Figure 1C*).

## Molecular mapping of the *L. vulgaris* nervous system

The central nervous system of *L. vulgaris* could be anatomically characterized using DAPI staining and were reinforced by HCR ISH experiments using broad neuronal markers. The anatomical characterization of brain structures is usually relying on the identification of neuropils, which are regions that have large numbers of axons and synapses but few cell bodies. Since the methods used here to describe gene expression are detecting RNA molecules that are mostly located in the cell bodies, usually surrounding nuclei, we briefly describe here the anatomy of the *L. vulgaris* nervous system in a nonexhaustive manner to facilitate the understanding of the results. The largest structure in the *L.*

*vulgaris* nervous system are the optic lobes. The cortex of the optic lobes is composed of an outer and inner plexiform layers and outer/inner granular layers. The rest of the optic lobe is referred to as the medulla. The suboesophageal mass and supraoesophagal mass make up the central part of the brain and contain several different lobes. Discussed in this article from anterior to posterior are the inferior buccal lobe, superior buccal lobe, frontal lobe, anterior basal lobe, precommissural lobe, vertical lobe, medial basal lobe, and dorsal basal lobe. The smaller peduncle lobe and olfactory lobe are posterior to the optic lobes. Note that the nervous system of *L. vulgaris* has several posterior lobes (posterior to the basal lobes) that are not shown on the confocal micrographs of this study as they have been sometimes lost during sample preparation after amputation of the head. A rough schematic of the arrangement of the different lobes (not at scale) can be found in *Figure 2—figure supplement 1*.

To search for neuronal subtypes in the nervous system, we looked specifically at the cell clusters that express neuronal makers (*Figure 2A*). We then assessed the expression patterns of neuronal markers using HCR ISH to gain better insights into the molecular identities of neuronal populations. Probes were generated for the genes *elav-like 1, elav-like 2, amyloid beta-binding-like,* and *tetraspanin-8,* which were all found to be expressed broadly across all neuronal clusters (*Figure 2B*). HCR ISH experiments for these probes confirmed their expression throughout the nervous system of *L. vulgaris,* but different genes showed specificities in their expression domains and expression levels. Some regions of the nervous system show co-expression of *elav-like 1* and *elav-like 2* (*Figure 2C*). Both genes are highly expressed in the optic lobes and in the neurons of the supraesophageal and suboesophageal mass. However, differences in the expression patterns of *elav-like 1* and *elav-like 2* can be observed: *elav-like 1* is strongly expressed in the medial part of the supraesophageal mass, including neurons of the dorsal basal lobe, vertical lobe, and subvertical lobe, as well as in the optic lobes in the medial portion of the medulla (*Figure 2C*, *Figure 2—figure supplement 2*). *Elav-like 2*, on the other hand, is expressed broadly in the lateral edges of the supraesophageal mass near the optic lobes and in the optic lobes. Interestingly, photoreceptors of the retina do not express *elav-like 1* but only *elav-like 2* (*Figure 2C*, *Figure 2—figure supplement 2*). The different expression domains are reflected by the scRNAseq data that shows, for instance, that *elav-like 2* expression level is higher than that of *elav-like 1* in Cl05, whereas the opposite can be observed in Cl07 (*Figure 2B*, *Figure 2—figure supplement 2*). Differences in the expression of these two ELAV homologs suggest that these genes could either serve different functions specific to the neuronal subpopulations in which they are expressed or that the different expression patterns reflect distinct developmental or differentiation states. The data shows that *elav-like 2*-expressing cells are present in the central and peripheral nervous system as well as in the lateral lips, which are shown to be connected to stem cells and neuronal differentiation later in the article (see *Figure 3*).

Expression of the gene *tetraspanin-8* was assessed and observed to be present in all cells throughout the brain, providing *tetraspanin-8* as a strongly expressed pan-neuronal marker. This high neural expression pattern suggests that *tetraspanin-8* is for the function of differentiated neurons in *L. vulgaris* providing an interesting candidate for the study of neuronal function in cephalopods. The pan-neuronal expression of *tetraspanin-8* allowed precise description of neuroanatomical regions of the *L. vulgaris* central nervous system (*Figure 2—figure supplement 3A–C*). Note that in many pictures shown in this study, DAPI appears absent in nervous tissue. This is the result of a lack of penetrability of DAPI inside the tissue but does not in any case signify the absence of nuclei in these structures.

Similarly, the expression of *amyloid beta-binding-like* is observed broadly throughout the entire brain, retina, and other components of the nervous system (*Figure 2C*). However, at higher magnification, it becomes evident that expression levels differ between cells. This is specifically apparent in the cortex layers of the optic lobe, where the cells of the inner granular layer express lower levels of *amyloid beta-binding-like* compared to *tetraspanin-8* (*Figure 2C*, *Figure 2—figure supplement 3D–G*). Low expression of *amyloid beta-binding-like* can also be observed in the olfactory organ (*Figure 2—figure supplement 3F*). Additionally, the scRNAseq data shows that the expression of *amyloid beta-binding-like protein* is lower in the reflectin[+] clusters (Cl20, Cl22, *Figure 1—figure supplement 2*).

The transcriptional repressor *scratch-2* was found to be highly expressed in two neuronal clusters (Cl05, Cl15) as well as in a reflectin[+] cluster (Cl22). HCR ISH expression analysis showed that this gene is mostly expressed in the medulla of the optic lobes and in the supraesophageal mass in two

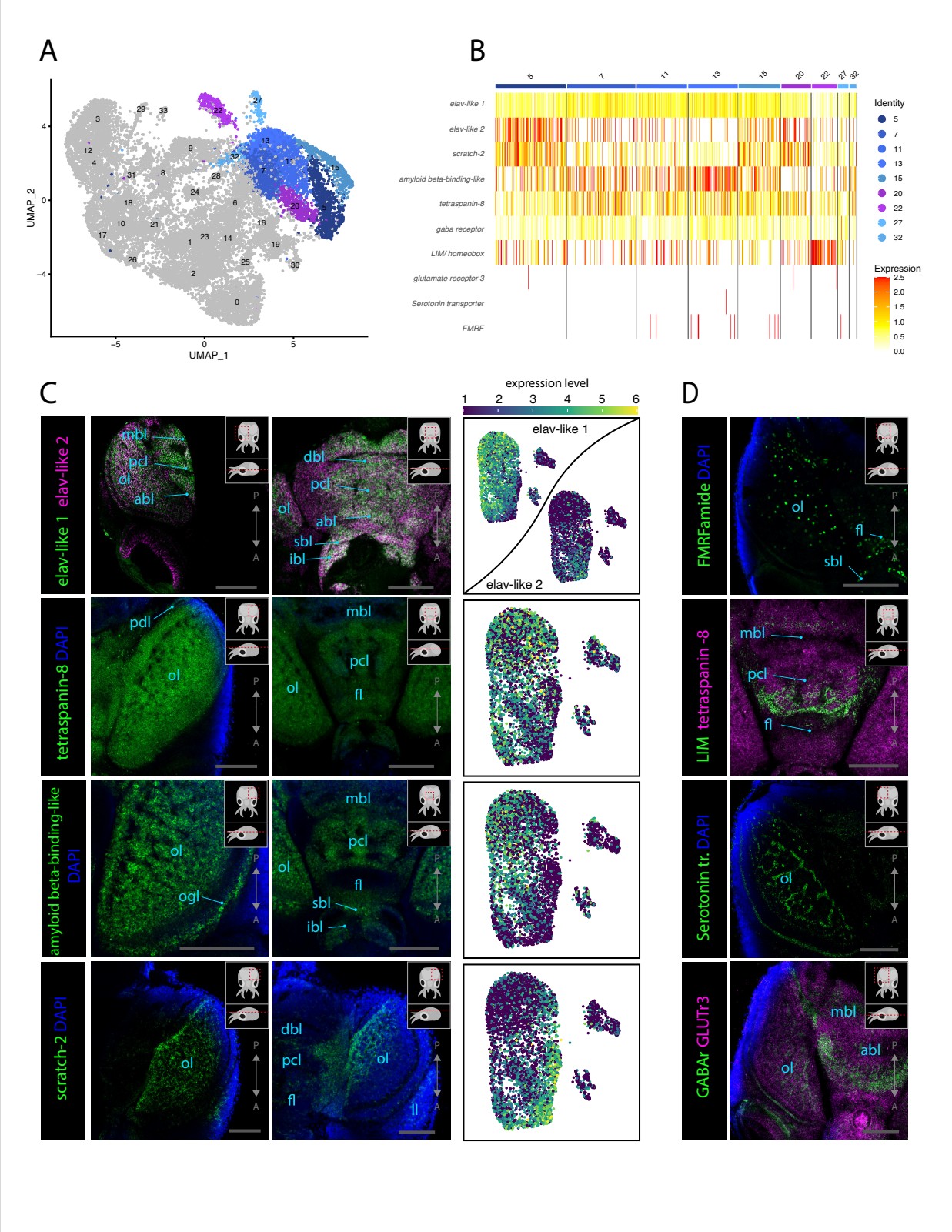

**Figure 2.** Molecular mapping of the *L. vulgaris* nervous system. (**A**) UMAP with clustered identified as neuronal or neuron-related highlighted. (**B**) Heatmap showing the expression of neuronal genes across each cell of the dataset. Cells are grouped by clusters. (**C**) Confocal micrographs showing mRNA expression detected by hybridization chain reaction (HCR) for broad neuronal markers identified in the scRNAseq data. The right-hand pictures are feature plots (UMAP) showing the expression of genes across the subclustered neuronal cells. Cartoons on the top right corner of each micrograph

*Figure 2 continued on next page*

*Figure 2 continued*

indicate the position of the image on a frontal view (top) and the plane of acquisition on sagittal view (bottom). Scale bars = 250 μM. (**D**) Confocal micrographs showing mRNA expression detected by HCR for additional neuronal markers; some of these markers were not differentially expressed in specific clusters. Note that DAPI staining in some cases has not penetrated the tissue, therefore explaining its absence from the optic lobes. Cartoons on the top-right corner of each picture indicate the position of the image on a frontal view (top) and the plane of acquisition on sagittal view (bottom). Scale bars = 250μM. (abl) anterior basal lobe, (dbl) dorsal basal lobe, (fl) frontal lobe, (ibl) inferior buccal lobe, (ll) lateral lip, (mbl) medial basal lobe, (ol) optic lobe, (Oo) olfactory organ, (ogl) outer granular layer, (pcl) precommissural lobe, (pdl) peduncle lobe, and (sbl) superior buccal lobe.

The online version of this article includes the following figure supplement(s) for figure 2:

**Figure supplement 1.** Rough schematic of the brain lobes of *Loligo vulgaris.*

**Figure supplement 2.** Expression patterns of *elav-like* 1 and *elav-like* 2.

**Figure supplement 3.** Expression patterns of *amyloid beta-binding-like* and *tetraspanin-8.*

**Figure supplement 4.** Expression pattern of *scratch-2.*

**Figure supplement 5.** Expression pattern of *fmrf.*

**Figure supplement 6.** Expression patterns of *lim homeobox* and *serotonin transporter.*

**Figure supplement 7.** Expression patterns of *glutamate receptor 3* and *gaba receptor.*

**Figure supplement 8.** (**A**) Expression of neurotransmitter and neuropeptide-related genes in the single-cell dataset.(**B**) Dotplots showing the average expression and percentage of cells expressing given genes in each cluster. Groups of genes were plotted separately to adapt the scale and improve readability of the plots.

domains lateral–ventral to the vertical lobe and anterior to the basal lobe that extends anteriorly toward the frontal lobes, as well as a stripe of cells connecting the two domains posterior to the vertical lobe (*Figure 2C*, *Figure 2—figure supplement 4*). The analysis of the expression of *scratch-2* in the scRNAseq dataset highlights that the cells highly expressing this gene are distinct from neurons expressing *elav-like 1*, *tetraspanin-8,* and *amyloid beta-binding-like* but show more transcriptional similarities with *elav-like 2*-expressing cells (*Figure 2C*).

*Fmrf* was shown to be expressed in some cells within the nervous system in agreement with previous anatomical studies (*Wollesen et al., 2010*; *Wollesen et al., 2012*). It is expressed in a region located between the dorsal basal lobe and the anterior basal lobe, surrounding the precommissural lobes, the peduncle lobe, the olfactory lobes, on the posterior part of the buccal mass and in the olfactory organ. Some cells expressing *fmrf* are also scattered throughout the optic lobes (*Figure 2D*, *Figure 2—figure supplement 5*).

The homeobox transcription factor *LIM homeobox* is expressed in the vertical lobe and subvertical lobe in a domain located between the dorsal basal lobes and the frontal lobes. It is also expressed around the peduncle lobe and in some cells at the surface of arms and tentacles (*Figure 2D*, *Figure 2—figure supplement 6*).

The expression of additional genes that were not identified as cell type specific in the scRNAseq but traditionally characterize neuronal populations based on their neurochemical modalities were assessed with HCR ISH. We identified a *serotonin transporter* gene that is expressed in the nervous system and more particularly in the medulla of the optic lobes. Low expression of this gene can be observed throughout the nervous system and indicates that the expression of the gene in the nervous system is not specific to an anatomical structure (*Figure 2D*). This transporter was, however, very highly expressed in the cornea of eyes (*Figure 2—figure supplement 6D–F*).

Two receptors for glutamate (*glutr3*) and GABA (*gaba-r*) were also tested with HCR ISH. The results show that a very large proportion of neurons express *glutr3* but *gaba-r* expression is mostly restricted to the medial part of the brain and a few cells of the optic lobes (*Figure 2D*). More specifically, *gaba-r* is expressed in the region posterior to the dorsal basal lobes, between the anterior basal lobes and the frontal lobes, in the peduncle lobes and the superior buccal lobes (*Figure 2—figure supplement 7*). *Gaba-r* appears to also be highly expressed in both neuronal and non-neural parts of the eyes but is absent from the optic lobes. Expression of *glutr3*, on the other hand, is observed all over the animals in neuronal tissues but also in epithelial and muscular tissues. This extremely broad expression pattern for *glutr3* indicates that this gene is likely to have other functions unrelated to neuronal communication. The genes involved in neurotransmission normally used to differentiate neuronal populations were expressed only in few cells in the single-cell dataset (*Figure 2—figure supplement 8*). This could

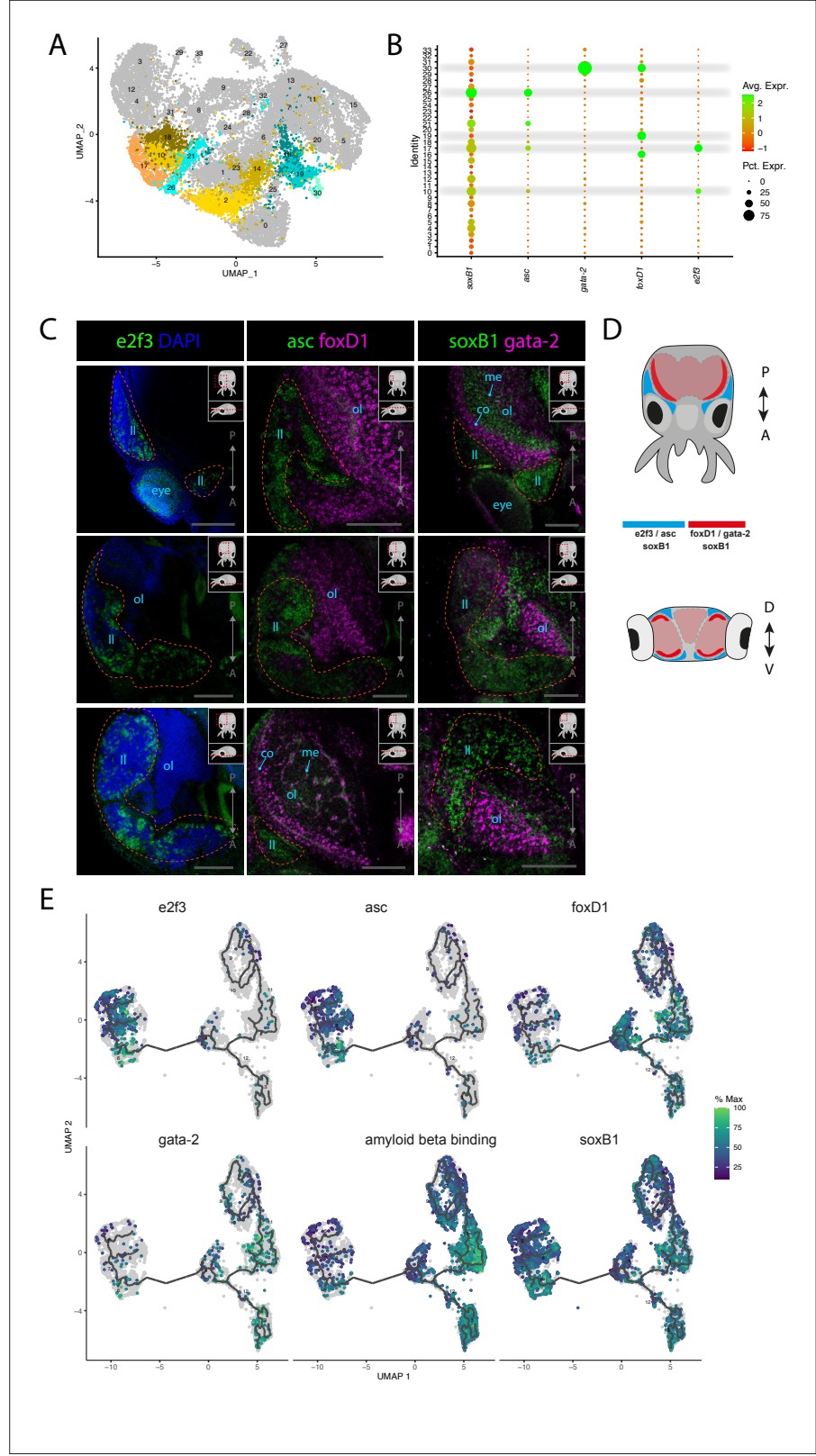

**Figure 3.** Stem cells and neuronal differentiation. (**A**) UMAP with clusters identified as either stem cell, Asc+, or FoxD+ highlighted. (**B**) Dot plot showing the average expression of marker genes and proportion of cells in each cluster that express these genes. (**C**) Confocal micrographs showing mRNA expression detected by hybridization chain reaction (HCR) for genes involved in stem cells and neuronal differentiation. Cartoons on the top-right

*Figure 3 continued on next page*

*Figure 3 continued*

corner of each picture indicate the position of the image on a frontal view (top) and the plane of acquisition on sagittal view (bottom). Red circles highlight the location of the lateral lips. Scale bars = 250µM. (co) cortex of the optic lobe, (eye) eye, (ll) lateral lip, (me) medulla of the optic lobe, and (ol) optic lobe. (**D**) Graphical representation summarizing the gene expression patterns of the genes assessed by HCR in situ hybridization (ISH) in the head of *L. vulgaris* with dorsal view (top) and transversal view (bottom). P/A, posterior/anterior; D/V, dorsal/ventral. (**E**) Plots showing the expression patterns of indicated genes across neurons and stem cells. The clusters and the trajectories are determined using monocle.

The online version of this article includes the following figure supplement(s) for figure 3:

**Figure supplement 1.** Expression pattern of *e2f3*.

**Figure supplement 2.** Expression patterns of *achaete-scute* and *foxD1*.

**Figure supplement 3.** Expression patterns of *soxB1* and *gata-2*.

**Figure supplement 4.** Monocle-based clustering and trajectory analysis.

**Figure supplement 5.** Trajectory inference using slingshot.

be caused by issues with the quality of the cells or a lack of mapping to the reference transcriptome. Possible reasons for these issues are discussed later in the article.

## Stem cells and neuronal differentiation

Single-cell sequencing makes it possible to identify cells not only based on their functional cell type identity but also based on their developmental stage. By identifying transcription factors that have been shown to be involved in cell type specification in previous studies and by following the expression of these genes across different cells, single-cell transcriptomics can provide insights into the different developmental lineages that specify cell types in *L. vulgaris*.

Several marker genes expressed in specific clusters suggest an involvement in proliferation and possess some conserved markers of neuronal differentiation. Two clusters (Cl02, Cl14, *Figure 3A*) do not express specific transcription factors that can be associated with developmental processes but show enrichment of genes involved in cellular proliferation and protein synthesis (*Figure 1—figure supplement 2*). In Cl10 and Cl17, the transcription factor *e2f3* is expressed together with other indicators of cellular proliferation, cellular growth, and nucleic acid synthesis such as histone proteins (*H3*), ribosomal proteins (*40s ribosomal protein S13, 40s ribosomal protein s19*), *pcna*, and DNA replication licensing factors (*mcm3, mcm4, mcm5*). The role of *e2f3* to promote cellular proliferation has been shown to be conserved between *D. melanogaster*, *Caenorhabditis elegans*. and mammalians (*Dynlacht et al., 1994*; *Attwooll et al., 2004*; *DeGregori and Johnson, 2006*). Because these clusters highly express proliferation markers but lack any distinctive features of differentiated cell types, we propose that they may be stem cells. To visualize the expression pattern of this gene, we performed HCR ISH experiments for *e2f3*. The results show that this gene is specifically expressed in the anatomical structure called the lateral lips, which have been described in *O. vulgaris* (*Figure 3C and D*, *Figure 3—figure supplement 1*, *Deryckere et al., 2021*). This suggests the presence and proliferation of stem cells or progenitors in the lateral lips of squids, consistently with the proposed neurogenic regions of *O. vulgaris*.

In Cl17, Cl21, and Cl26, the transcription factor *achaete-scute* (*asc*), which has a well-studied role in the regulation of nervous system development in *D. melanogaster*, is expressed together with similar proliferation markers as indicated previously (*Cabrera et al., 1987*; *Romani et al., 1987*; *Skeath and Carroll, 1992*). This is particularly interesting to assess the possible conservation of this gene's function in nervous system development. The expression pattern revealed by HCR ISH showed a strong similarity with the expression of *e2f3* in the lateral lips (*Figure 3C and D*, *Figure 3—figure supplement 2*). This suggests that neuronal progenitors and stem cells are co-located in the lateral lip where they proliferate. Frequent co-expression with *soxB1* in these clusters further supports their neuronal fate because of the conserved involvement of *soxB1* genes in neuronal development in very distantly related species such as *C. elegans* and mammals as well as in cephalopods (*Arsic et al., 1998*; *Miyagi et al., 2009*; *Alqadah et al., 2015*; *Focareta and Cole, 2016*).

We next analyzed the expression of *soxB1*, which revealed enriched expression in the lateral lips, similarly to *e2f3* and *asc*. Interestingly, the pattern of expression of *soxB1* is broader than *asc* and is

also present in most lobes of the brain (*Figure 3—figure supplement 3*). Together, the overlapping expression patterns of *e2f3, asc*, and *soxB1* in domains associated with cephalopod neurogenesis strongly suggest that stem cells and neuronal progenitors are present the lateral lips of *L. vulgaris*, suggesting that these cells are likely migrating to the nervous system at later differentiation stages. In agreement with a pro-neural function, these cells do not yet express markers of terminally differentiated neurons. It was previously shown that *soxB1* is expressed throughout late embryogenesis in the developing eye of *D. pealeii* (*Napoli et al., 2022*). We indeed found comparable expression of *soxB1* in the developing eye in *L. vulgaris* pre-hatchling and further detected expression in the developing olfactory organ, suggesting a broader role of *soxB1* in sensory organ development or function (*Figure 3—figure supplement 3*).

Cl16, Cl19, and Cl30 are characterized by the expression of the forkhead box transcription factor *foxD1*. FoxD genes have been documented to be expressed in the nervous systems of *Mus musculus*, *Xenopus laevi*, *C. elegans*, and *D. melanogaster* and to play a role in the development of neuronal cell types although not exclusively (*Herrera et al., 2004*; *Polevoy et al., 2017*; *Newman et al., 2018*; *Janssen et al., 2022*). HCR ISH assessment of the expression of *foxD1* showed that these cells are strongly expressed in the cortex of the optic lobes and in the medial basal and anterior basal lobes (*Figure 3C*). The expression of *foxD1* is very strong in the cortex of the optic lobes and in the medial and anterior basal lobes but is also sparsely located in the medulla (*Figure 3—figure supplement 2*). The expression pattern of *foxD1* indicated that these cell types are present in the nervous system and could therefore be types of neurons or neuronal progenitors. Cl30 also expressed a GATA zinc finger transcription factor (*gata-2*) together with some markers typically associated with blood cells (*hemocyte protein, immunoglobulin, soma-ferritin, apolipophorin*). The expression pattern of *gata-2* is very similar to *foxD1* and is also strongly expressed in the cortex of the optic lobes (*Figure 3C*, *Figure 3—figure supplement 3*). *Gata-2* expression can also be found at the surface of the arms and in the suckers. The HCR ISH experiments with these different genes showed that the expression of *e2f3* and *asc* is limited to the proliferating cells located in the lateral lips, whereas the expression of *foxD1* and *gata-2* is not found in the lateral lips but inside nervous tissue. The expression pattern of *soxB1* revealed that it is present both in the lateral lips and the nervous tissue of the optic lobes (*Figure 3D*).

To assess the possible developmental lineages that could link these different clusters together, we performed trajectory analysis on a subset of cells. We selected all the cells from the stem cell clusters (Cl02, Cl10, Cl14, Cl17), the neuronal clusters including photoreceptors and presumed sensory cells (Cl05, Cl07, Cl11, Cl13, Cl27, Cl32), the foxD+ clusters (Cl16, Cl19, Cl30), and the asc+ clusters (Cl21, Cl26). These cells were used for subclustering and subsequent trajectory analysis using Monocle3 (*Trapnell et al., 2014*; *Qiu et al., 2017b*; *Qiu et al., 2017a*). As monocle operates with its own clustering algorithm, we fist verified that the markers tested previously were still clustered together and then proceed to calculate trajectories. We selected the *e2f3* expressing cluster as a starting point for the trajectories and the results indicated that following the proliferative stage where cells express *e2F3* and *asc,* cell trajectories go through a stage where *foxD1* and *gata-2* are strongly expressed (*Figure 3E*). After this stage, we see increased expression of the markers of differentiated neurons *amyloid beta-binding-like* (*Figure 3E*). The expression of *soxB1* along the trajectory axis appears to be present in the early stages as well as in the differentiated neurons, consistently with the expression patterns previously observed by HCR ISH (*Figure 3C and E*). The expression level of *soxB1* is, however, very low in the *foxD1*-expressing stage. Additional markers could be plotted along the trajectory axis: the proliferation markers *pcna* and *histone H3* and the transcription factor *elf-2* show similar expression as *e2f3* and *asc* in the early stages of differentiation (*Figure 3—figure supplement 4*). The hypothetical intermediate stage characterized by the expression of *foxD1* and *gata-2* could also be correlated with the expression of the transcription factors *ets4* and *hes-4* as well as *chemotrypsin* (*Figure 3—figure supplement 4*). *Elav-like 1*, *elav-like 2,* and *synaptogyrin-1* are highly expressed in the differentiated, and although they are present across all clusters, the average expression increases along the differentiation axis (*Figure 3—figure supplement 4*).

To support the results shown by the monocle 3 trajectory inference, we performed a similar analysis with slingshot (*Street et al., 2018*). The same cells as for the monocle analysis were selected and re-clustered. The results obtained with slingshot were very similar to those obtained with monocle and showed a trajectory that supports foxD1+ cells as a possible intermediate stage between the proliferative stage where *e2f3* and *asc* are expressed and the differentiated neurons expressing *amyloid*

*beta-binding-like* and higher levels of *elav-like 1* and *elav-like 2* (**Figure 3—figure supplement 5**). The calculation of trajectories using these two separate methods provides insights into the developmental relationship between these different identified cell types and provides new hypotheses for the study of neurogenesis of cephalopods.

## Sensory cells

Vision is a crucial sense for various behaviors of the squids as it is needed for navigating efficiently in open waters in the wild. The eyes of squids are very large and the optic lobes, where visual information is processed and integrated, take up a significant part of the overall volume of the brain. The phototransduction cascade that allows light stimuli to be transformed into an electrical signal in the neurons is well conserved among cephalopods, and we were able to clearly identify photoreceptors in the scRNAseq data (**Davies et al., 1996**; **Monk et al., 1996**; **Arendt, 2003**). Many important components of the phototransduction cascade are highly expressed in Cl32 such as *trp-like*, *retinochrome*, *arrestin*, *phospholipase C*, and *rhodopsin*. To verify the expression of members of the phototransduction cascade in the photoreceptors, HCR ISH probes for *trp-like* and *rhodopsin* were selected and tested. Both genes showed striking specificity and are expressed in the photoreceptors of the retina (**Figure 4A**, **Figure 4—figure supplement 1**). The olfactory organs of *L. vulgaris* are located on the ventral surface of the head (see **Figure 1C**). The genes involved in the function of these sensory organs remain unknown and thus no cell type-specific markers were readily available to identify chemosensory cells of the olfactory organ. However, through screening of some cluster markers of interest, we could observe that the olfactory organ strongly expresses FMRF-related neuropeptides (*fmrf*). Interestingly, the pattern of expression of *fmrf* in the olfactory organ (**Figure 4B**, **Figure 2—figure supplement 4**) strongly resembles that of *reflectin-8* (**Figure 4B**, **Figure 4—figure supplement 2**), a gene typically involved in the function of iridophores, which are a specialized subclass of chromatophores. FMRFamide has been previously shown to be involved in the control of chromatophores in squids (**Loi and Tublitz, 2000**; **Sweedler et al., 2000**; **Satpute-Krishnan et al., 2006**; **Mobley et al., 2008**). Additionally, we observed expression of *elav-like 1* in neurons of the epidermal lines, stripes of cells located on the dorsal epidermis of cephalopods that have been proposed to have a function similar to the lateral lines of fish (**Figure 4B**, **Figure 1—figure supplement 1**; **Budelmann and Bleckmann, 1988**). Epidermal lines were suggested to be mechanosensory cells, used to sense water currents in the cuttlefish *Sepia officinalis* and in the squid *Lolliguncula brevis* (**Komak et al., 2005**). The expression of *elav-like 1* in this structure in *L. vulgaris* is consistent with their proposed mechanosensory function other cephalopods.

We observed expression of *cilia-associated protein* (Cl27) at the surface of the arms (**Figure 4B**). The scRNAseq data revealed that the cells of this cluster frequently co-express the synaptic markers *rab3* and *synaptogyrin-1*. Because of the expression of *cilia-associated protein* at the surface of the arms, co-expressed with neuronal markers and additional cilia/flagella markers such as *kinectin-like* and *dynein light chain*, we propose that these cells could be specific mechanoreceptors of the arms of *L. vulgaris*. The expression of marker genes used to identify sensory cell types is summarized in **Figure 4C**. The sensory markers that were identified were plotted onto the neuronal subclustered previously used in **Figure 2**. The results highlight the co-expression of neuronal markers and sensory markers, therefore supporting their sensory identity (**Figure 4D**).

## Epidermis, connective tissue, and cephalopod-specific cell types

The epidermis of cephalopods is covered with pigmented cells organized in precise patterns. Additionally, the cephalopod-specific chromatophores, which are colored or iridescent, have been shown to be modulated by the nervous system (**Dubas et al., 1986**; **Wardill et al., 2012**; **Gonzalez-Bellido et al., 2014**). Cl08 and Cl09 could be identified as epidermal cells from the scRNAseq data because of the consistent expression of epidermal growth factor EGF-like. In Cl08 and Cl09, *egf-like* was co-expressed with an aminopeptidase (*xaa-pro aminopeptidase*) and *feeding circuit activating peptide-like*, a neuropeptide that was shown to be involved in the feeding circuit in the nervous system of *Aplysia californica* (**Sweedler et al., 2002**). Cl08 expressed the markers *tyrosinase* and *tryptophan 2–3 dioxygenase*, two enzymes involved in melanin synthesis pathway (**Takeuchi et al., 2005**; **Crawford et al., 2020**). HCR ISH experiments confirmed that *egf-like* is expressed in several cells across the epidermis of the animal (**Figure 5B**, **Figure 5—figure supplement 2**). Cl31 was characterized as epithelial due

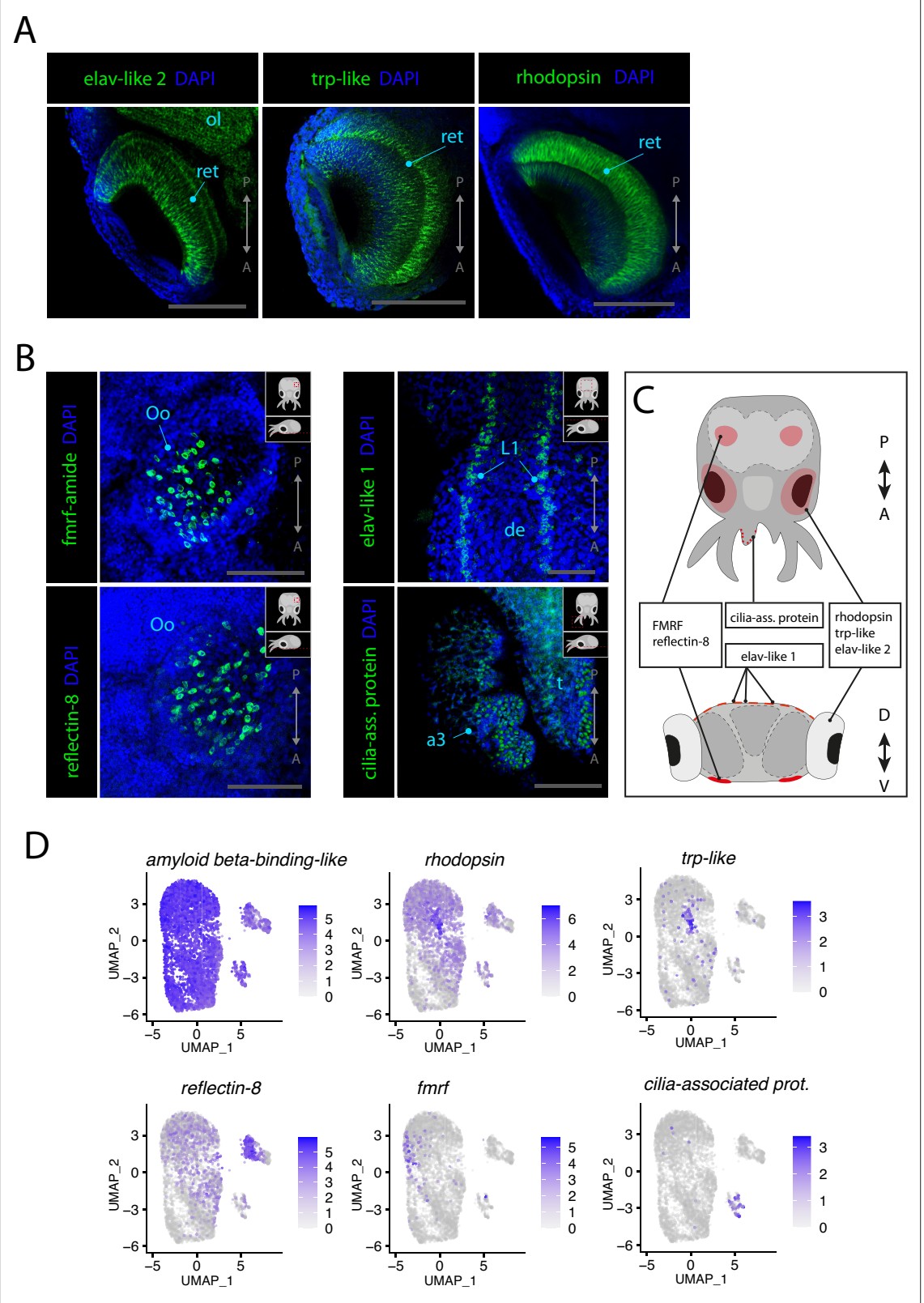

**Figure 4.** Sensory cells. (**A**) Confocal micrographs showing mRNA expression detected by hybridization chain reaction (HCR) for genes expressed in photoreceptors. Scale bars = 250μM. (**B**) Confocal micrographs showing mRNA expression detected by HCR for genes expressed in chemosensory organs and mechanosensory cells. Cartoons on the top-right corner of each picture indicate the position of the image on a frontal view (top) and the plane of acquisition on sagittal view (bottom). Scale bars = 250 μM. (**C**) Graphical representation summarizing the expression of identified genes

*Figure 4 continued on next page*

*Figure 4 continued*

in specific sensory organs and regions. Dorsal view (top) and transversal view (bottom). P/A, posterior/anterior; D/V, dorsal/ventral. (**D**) UMAP of the subclustered neuronal subset with the plotted expression level of the different identified sensory markers. (a3) arm pair 3, (de) dorsal epidermis, (L1), epidermal lines 1, (ol) optic lobe, (Oo) olfactory organ, (ret) retina.

The online version of this article includes the following figure supplement(s) for figure 4:

**Figure supplement 1.** Expression patterns of *rhodopsin-1* and *trp-like*.

**Figure supplement 2.** Expression pattern of *reflectin-8*.

to the very specific expression of *epithelial splicing regulatory protein* (*esrp*). HCR ISH for this gene revealed that it is expressed specifically on the surface of the arms, on the epithelium covering the surface of the eyes and in cells of the epidermis, similarly to *egf-like*. *Esrp* is also expressed at a lower level across the surface of the epidermis (*Figure 5B*, *Figure 5—figure supplement 1*). Cl03, Cl04, and Cl12 showed specific enrichment for *papilin*. Studies on the role of *papilin* in *D. melanogaster* revealed that it is a secreted extracellular protein that acts as a metalloproteinase and is important for embryonic development (*Kramerova et al., 2000*; *Kramerova et al., 2003*). In *L. vulgaris* as well as other cephalopods, the presence of this protein is documented but its function is unknown. These clustered also showed elevated expression of *gluthatione S-transferase,* indicating a possible role of these cells in metabolizing reactive oxygen species. The location of *papilin*-expressing cells was assessed with HCR ISH, and we observed that this gene is expressed in a very particular set of cells of the skin that form characteristic rings around larger epidermal cells (*Figure 5B*). Although *papilin* is expressed everywhere on the epidermis of the head of *L. vulgaris,* it is absent from the surface of the olfactory organ (*Figure 5—figure supplement 2*). This suggests that the layering of the olfactory organ is composed of different cell types than the rest of the epidermis. Cl28 is characterized by the expression of a zinc metalloproteinase *nas 14-like* of unknown function. Due to the co-expression *of collagen-related* genes, *myosins.* and *actins*, these cells were assigned to connective tissue. HCR ISH experiments showed expression of *nas 14-like* throughout the body in large cells with extended processes (*Figure 5C*). The branched morphology of these cells is similar to fibroblasts. *Nas-14* is expressed predominantly in the ventral epidermis but is also observed in the dorsal epidermis, the arms, and tentacles and is heavily expressed on the surface of the funnel (*Figure 5—figure supplement 3*). We analyzed the expression of *reflectin-8,* one of the several *reflectins* expressed in Cl20 and Cl22. However, in addition to having been shown to be also expressed in the olfactory organ (see *Figure 4*), HCR ISH for this *reflectin* showed broad expression at the surface of the epidermis but also inside the optic lobes and around the anterior basal lobes, superior buccal lobes, and peduncle lobes (*Figure 5C*, *Figure 4—figure supplement 2*). The expression pattern of *reflectin-8* in the nervous system suggests that the involvement of this protein may go beyond its ability to modulate light reflection and could play a role inside the nervous system itself for the modulation of chromatophores and iridophores as is suggested in previous literature (*Demski, 1992*). This is also supported by the co-expression of reflectins and the neuronal markers *scratch-2*, *elav-like 1*, *elav-like 2*, and *rhodopsin-1* in Cl20 and the expression of *LIM homeobox* in Cl22. A *mesencephalic astrocyte-derived neurotrophic factor* (*madnf*) is expressed in Cl23, Cl27, and Cl31, which were characterized as connective tissue, mechanosensory cells, and epidermis, respectively. HCR ISH experiments for this gene surprisingly show an expression pattern very similar to *reflectin-8*. The results showed that this gene is expressed throughout the nervous system, including in the optic lobes in the medial portion of medulla (*Figure 5—figure supplement 4*). While *madnf*-expressing cells of Cl27 co-express neuronal markers, the cells of Cl23 and Cl31 do not. Given the observed expression pattern of this gene, it is possible that these cells are a non-neuronal cell type of the nervous system and could be a type of glial cells. Suckers contain a specific cell type that makes up the hard structure of the sucker ring teeth, which are used for holding on to their prey. The sucker ring teeth consist of the protein suckerin, for which we found gene expression in Cl29, which is associated with epidermal tissues. The cells of the suckers expressed many uncharacterized genes, indicating that these cell types are highly specialized in cephalopods. Moreover, we noted co-expression of *myeloperoxidase-like,* a gene involved in antimicrobial activity. It has been shown that *O. vulgaris* suckers express peptides that display structural similarities with antimicrobial α-peptides (AMPs) and exhibited a significant role in antimicrobial activities (*Maselli et al., 2020*). It is likely that octopuses and other cephalopods exploit secreted AMPs

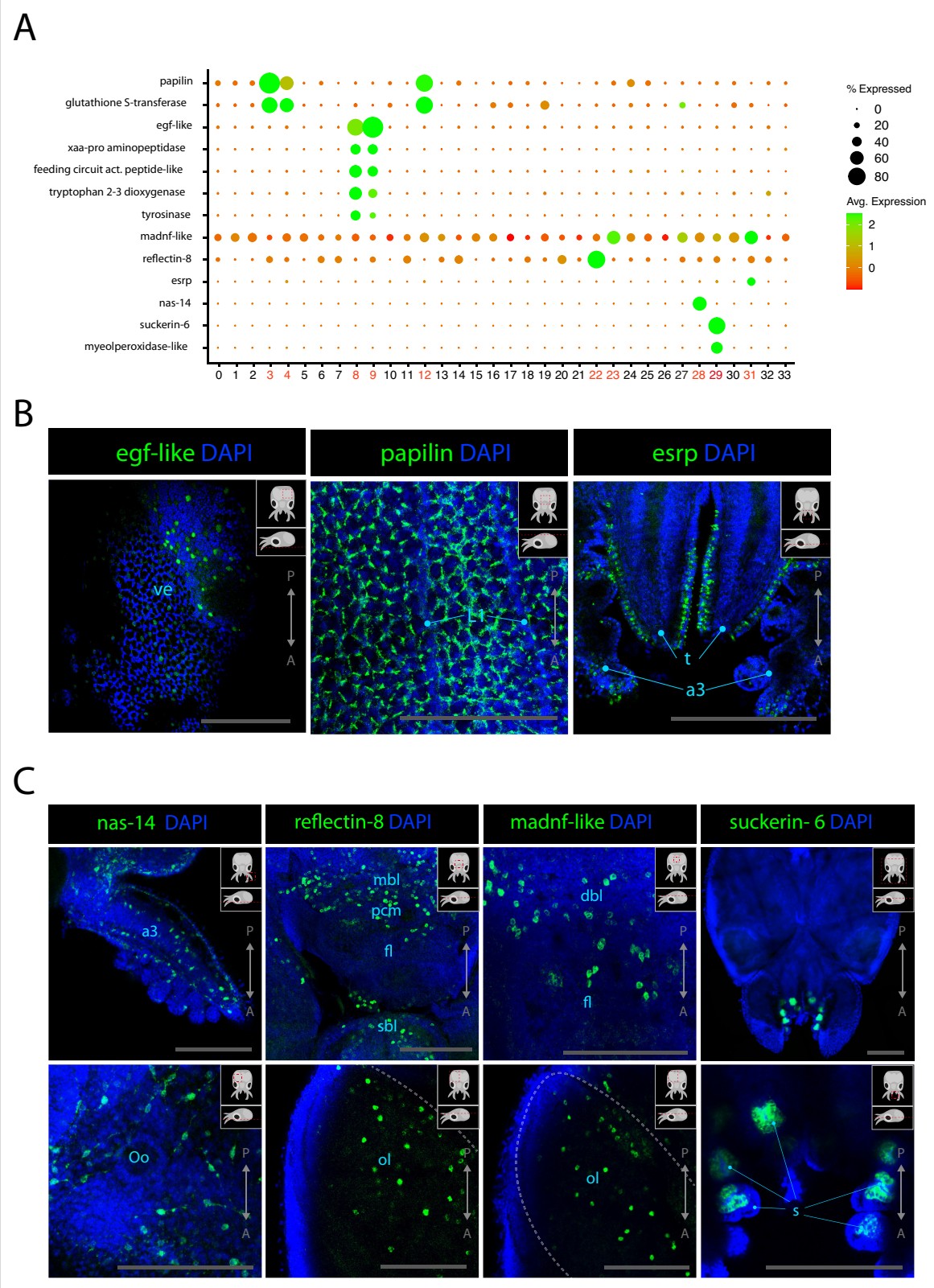

**Figure 5.** Epidermis, connective tissue, and cephalopod-specific cell types. (**A**) Dot plot showing the average expression of marker genes of interest and proportion of cells in each cluster that express these genes. Clusters identified as epidermal, connective tissue, or reflectin+ are labeled in red. (**B**) Confocal micrographs showing mRNA expression detected by hybridization chain reaction (HCR) for genes involved in epidermis development or function. Cartoons on the top-right corner of each picture indicate the position of the image on a frontal view (top) and the plane of acquisition on

*Figure 5 continued on next page*

*Figure 5 continued*

sagittal view (bottom). Scale bars = 250µM. (**C**) Confocal micrographs showing mRNA expression detected by HCR for genes expressed in connective tissue or associated with iridophores and suckers. Cartoons on the top-right corner of each picture indicate the position of the image on a frontal view (top) and the plane of acquisition on sagittal view (bottom). Scale bars = 250 µM. (a3) arm 3, (dbl) dorsal basal lobe, (fl) frontal lobe, (L1) epidermal lines 1, (mbl) medial basal lobe, (ol) optic lobe, (Oo) olfactory organ, (s) suckers, (sbl) superior buccal lobe, (t) tentacles, (ve) ventral epidermis.

The online version of this article includes the following figure supplement(s) for figure 5:

**Figure supplement 1.** Expression patterns of the *epithelial splicing regulatory protein* and *cilia-associated protein* genes.

**Figure supplement 2.** Expression patterns of *papilin* and *egf-like*.

**Figure supplement 3.** Expression patterns of *nas-14*.

**Figure supplement 4.** Expression pattern of *mesencephalic astrocyte derived neurotrophic factor*.

as part of an innate defense mechanism, similarly to other aquatic animals (*Thøgersen et al., 1992*; *Derby, 2014*; *Falanga et al., 2016*; *Gogineni and Hamann, 2018*). HCR ISH experiments showed that the expression of *suckerin-6* is specific to a cup-shaped group of cells in the inner sucker, likely becoming cells producing the sucker ring teeth (*Figure 5C*).

## Discussion

We here provide molecular insights into the cell types present in the head of *L. vulgaris* by combining two state-of-the-art techniques HCR ISH and scRNAseq. The combination of these two techniques allowed us to provide insights into the different cell types that make up the *L. vulgaris* head, into the process of nervous system development and the identity of neuronal subtypes. The data produced for this article enabled the identification of 34 cell clusters that we used as a correlate for cell type diversity. Although the data can provide a broad overview of cell diversity, we note that few cell types that would be expected in the head of cephalopods such as components of the vascular and digestive systems could not be confidently identified in this study. The apparent lack of resolution in the clusters and their low number is likely to be in large part caused by the lack of completeness of the reference transcriptome, which resulted in a lower mapping rate of the single-cell reads. It has been shown in other cephalopods that having a high-quality genome largely contributes to improving cluster resolution (*Gavriouchkina et al., 2022*; *Songco-Casey et al., 2022*; *Styfhals et al., 2022*). Because of the lack of a reference genome for *L. vulgaris,* a non-redundant transcriptome was used as a mapping reference in this study, which has shown to be sufficient for identifying cell types and describing the expression some of their characteristic marker genes but is however not sufficient to affirm full coverage of cell type diversity of this organism. The use of a transcriptome instead of a genome as a reference has been previously shown to enable molecular description of cell types but can lack in resolution (*Duruz et al., 2021*). Additionally, it is likely that the transcriptome assembly could be improved by obtaining more RNA from *L. vulgaris* by including different embryonic stages and adult stages. We are convinced that re-running this analysis with a better reference would reinforce our analysis and provide better cell type resolution. In our analysis of the nervous system, we were able to recognize broad neuronal markers that are expressed throughout the nervous system while showing different levels of expression in predicted neuronal cell types. These newly identified neuronal markers will provide a new perspective to study the characteristics of the nervous systems of cephalopods. Particularly, the genes *tetraspanin-8* and *amyloid beta-binding-like* show strikingly elevated expression pan-neuronally. Besides the identified sensory cells, the homogeneity of the neuronal clusters was rather surprising. Different neuronal subpopulations were expected to be identified by the expression of genes neurotransmitter synthesis pathway to distinguish neurons by their chemical modalities. Such fine distinctions between neuronal populations could be found in the nervous systems of *E. berryi* and *O. vulgaris* (*Gavriouchkina et al., 2022*; *Songco-Casey et al., 2022*; *Styfhals et al., 2022*). It is likely that this lack of resolution is due to missing genes in our reference transcriptome, which would classically contribute to distinguishing neuronal populations. The analysis of the expression of two highly expressed members of the ELAV family elav-like 1 and elav-like 2 revealed that although they can be found in most lobes of the nervous system, their expression levels differ in different clusters. Previous studies of ELAV proteins in *S. officinalis* revealed that the gene Sof-Elav1 was expressed in all the lobes of the nervous system but also during early embryogenesis while the other gene Sof-Elav2 was not

specific to neurons (**Buresi et al., 2013**). In *O. bimaculoides,* a single neuronal ELAV protein was identified and was observed to be also expressed during neurogenesis and could be used as a neuronal marker to study nervous system development during embryogenesis, similarly to *Sepia* (**Shigeno et al., 2015**). It is interesting to note that contrarily to what was described in cuttlefish and octopus we show two cell type-specific elav genes, which are both expressed in neurons. This challenges the hypothesis of a duplication of the elav gene during lophotrochozoan evolution proposed after the observation of two elav genes in *Capitella* sp.*,* one of which is expressed in neurons while the other is expressed mainly in the mesoderm (**Meyer and Seaver, 2009**). The identification of two neuronal elav genes in *L. vulgaris* raises the question of whether they are orthologous to the genes studies in *Sepia* and *Octopus* but their function is different. Alternatively, these two neuronal elav genes could the result of another duplication of the neuronal elav. Such questions would need to be further investigated by more thoroughly studying the sequence homology of these genes and reconstruct their phylogeny. We identified key transcription factors involved in stem cells and development. The expression of these genes likely reflects different states of neuronal differentiation and suggests a migration of neuronal progenitors towards the nervous system. We could show that proliferating cells that express of the transcription factor *e2f3* are located in the lateral lips near the nervous system. We interestingly observed the expression of *asc* in the same region, together with several proliferation markers. Extensive study of neurogenesis in *D. melanogaster* has shown that the proneural complex *asc* can determine the neuronal fate of quiescent multipotent ectodermal cells (**Campuzano and Modolell, 1992**; **Skeath and Doe, 1996**). In vertebrates, on the other hand, *asc* is expressed in cells that have already undergone neuronal specification but are self-renewing (**Bertrand et al., 2002**; **Soares et al., 2022**). Studies of neurogenesis in the annelid *Capitella teleta* have shown that, similarly to vetrebrates, proneural genes such as *asc* are expressed in self-renewing neuronal progenitors (**Sur et al., 2020**). Our data suggests that *asc* is also expressed in proliferating cells, similiarly to vertrebrates and *C. teleta* but unlike *Drosophila*. This is consistent with documented co-expression of *asc* and the mitotic marker phosphor-histone H3 in *O. vulgaris* (**Deryckere et al., 2021**). The observation of *soxB1* expression together with *asc* strengthened their characterization as neuronal progenitors due to the documented role of this gene in neuronal differentiation (**Miyagi et al., 2009**). However, strong soxB1 expression in differentiated neurons has been shown in the gastropod *Patella vulgate* and in the cephalopods *S. officinalis* and *O. vulgaris* (**Le Gouar et al., 2004Focareta and Cole, 2016**; **Deryckere et al., 2021**). In *L. vulgaris,* we observe similar results where *soxB1* is expressed in the lateral lips together with *asc* but is also expressed in differentiated neurons. This was consistently shown by the HCR ISH experiments as well as the single-cell analysis. Therefore, the hypothesis proposed for the action of *soxB1* in the early stages as well as the later stages of neuronal differentiation in *O. vulgaris* is consistent with our data (**Deryckere et al., 2021**). The unexpected expression of *foxD1* in the nervous system of *L. vulgaris,* together with existing evidence of the role of this gene in neuronal development, motivated the grouping of the foxD1[+] clusters with other clusters characterized as neuronal progenitors for further analysis (**Herrera et al., 2004**; **Polevoy et al., 2017**; **Newman et al., 2018**; **Janssen et al., 2022**). Our trajectory inference analysis using monocle3 and slingshot showed that *foxD1* is expressed in cells that could represent an intermediate neuronal differentiation stage following the expression of *asc* but before the expression of *amyloid beta-binding-like* and the second peak of *soxB1* expression. Such a role has been proposed in *O. vulgaris* in the cells expressing *neuroD* that are expressed in a gradient between the lateral lip and the central nervous system, which may indicate a pattern of migration of differentiating neurons (**Deryckere et al., 2021**). We could, however, not correlate these results with our hypothesis of an intermediate *foxD1*-expressing stage because *neuroD* expression could not be detected in specific clusters in our single-cell dataset. Although the trajectories we calculated point towards a role of FoxD1 in neuronal differentiation, the relatively poor cluster resolution and the lack of cells along the trajectory axes shown in *Figure 3* suggest that this hypothesis would require further investigation that should include a more complete set of genes that have been involved with cephalopod neurogenesis. In our analysis of sensory cells, we provide valuable information regarding the different sensory systems of *L. vulgaris* and could identify photoreceptors and additionally describe newly identified markers of the olfactory organ and mechanosensory cells. The sensory cells of cephalopods have been extensively studied and are known to be very complex and can have multiple sensory modalities (**van Giesen et al., 2020**; **Buresch et al., 2022**). In *O. bimaculoides* and *S. officinalis,* several chemosensory receptors could be identified and

characterized (*van Giesen et al., 2020*; *Andouche et al., 2021*). In our analysis of the *L. vulgaris* head, we were however not able to identify putative chemosensory genes, likely due to the absence of characteristic receptors in our transcriptome. The chemosensory function of cells analyzed here was only inferred because of their expression in the olfactory organ. A full analysis of chemosensory genes in *L. vulgaris* would be required to be able to truly describe the diversity of their sensory cell types. Finally, we further characterize the highly specialized epidermis of *L. vulgaris* and cephalopod-specific cell types such as suckers or iridophores. The expression of a few genes facilitated the characterization of some of these cell types as exemplified by the expression of *egf-like* in epidermal cells, *suckerin-6* in suckers, and *reflectin-8* in iridophores. However, many cluster-specific genes could not be assigned to a specific function. The abundance of papilin-expressing cells on the surface of the epidermis is striking and had not been previously documented. The gene *madnf* is known to play a role in the maturation and maintenance of dopaminergic neurons in *D. melanogaster* and *C. elegans* (*Palgi et al., 2009*; *Richman et al., 2018*). We found discrepancies between the observation of *madnf* in non-neuronal clusters and in a small cluster of mechanosensory cells and the expression patterns shown by HCR ISH, where expression is found in different areas of the brain, optic lobes, and olfactory organs. Similarly, the expression of *reflectin-8* is not limited to the chromatophores but is also in the nervous system. The expression of these non-neuronal makers inside the nervous system results highlights the importance of considering non-neuronal cell types, even when aiming to answer neurobiological questions. Many cell types remain unidentified, and further characterization of these cells and the function of the genes that they differentially express could advance our understanding of cephalopod biology. The combination of high-resolution imaging rendered possible by HCR in conjunction with the data provided by single-cell transcriptomics enabled us to extend our understanding of the cell types of cephalopods. The experimental approach is efficient to identify and explore conserved sets of genes conferring cell type identity in these animals while comparing cell type markers across different clades. On the other hand, the accurate anatomical description gained with HCR ISH allowed us to find previously undescribed cell type-specific genes by identifying the anatomical features in which they are expressed, enabling the exploration not only of conserved cell types but also novel ones. By applying single-cell RNA sequencing in a member of this fascinating clade of mollusks and by correlating this data with high-resolution mRNA imaging techniques, we can further understand the main molecular features that drive cell type identity in *L. vulgaris*. The analysis of the expression of selected genes of interest can provide insight into their function, which can be a starting point for future investigations of cephalopod cell types. Further studies of the nervous system of squids and the different cell types that compose them could help fill important gaps in our understanding of the evolution of the nervous system and could provide insights into how brains complexified independently in very distant clades of animals. By comparing the molecular features of neuronal cell types of cephalopods with other species, we could better understand the evolutionary history of the unique brains of cephalopods and identify what confers them such unique capabilities. This is a significant step in the study of convergent evolution of brains and provides a valuable resource for future research on cephalopods.

## Materials and methods

### *L. vulgaris* embryo collection and maintenance

*L. vulgaris* eggs were collected during a sampling project of the European Marine Biological Resource Center (EMBRC) France at the Bay of Morlaix. The eggs were then incubated in a closed artificial seawater system (Aquarium Red Sea system reefer deluxe 170 Black). The salinity was between 34 and 36 particles per thousand (ppt), and pH was between 7.80 and 8.40. A 12 hr light/dark cycle was set with a crepuscular light and a dawn light before the day and the dark cycle; the used lamps were the Maxspect éclairage LED de la series RSX 100W. The eggs were suspended per bundle in a plastic structure made with PVC. Each bundle entailed between 15 and 30 arms. Each arm contained between 100 and 150 embryos. Great care was allocated to guarantee egg oxygenation; according to *Le Gouar et al., 2004*; *Focareta and Cole, 2016*; *Deryckere et al., 2021*, the oxygen consumption of one egg mass is twice the consumption of an adult animal of mean size. Therefore, two additional bubblers with air diffusers were added (Tetra APS 100 and Tetratec AS45). The stage of the animals was assessed based on the description of embryonic

stages in *Loligo paelii* (*Arnold, 1965*). As the temperature was lowered to slow down development, the stage 28–30 animals (pre-hatchlings) were determined based on the presence of chromatophores but lack of visible ink sac and the size of the yolk sack approximately equivalent to the size of the head (*Arnold, 1965*).

## RNA extraction

Ten embryos were collected into QIAzol Lysis Reagent (QIAGEN, Cat# 79306), homogenized, and centrifuged. RNA was isolated with chloroform and precipitated with isopropanol at –20°C. After washing with ethanol, the RNA was resolved in DEPC-treated water.

## Transcriptome assembly

The quantity and quality of the extracted RNA were assessed using a Thermo Fisher Scientific fluorometer (Qubit 4.0) with the Qubit RNA BR Assay Kit (Thermo Fisher Scientific, Q10211) and an Advanced Analytical Fragment Analyzer System using a Fragment Analyzer RNA Kit (Agilent, DNF-471), respectively. The RNA was also tested by spectrophotometry using a Denovix DS-11 FX Spectrophotometer/Fluorometer to assess the purity of the RNA. Once all quality control tests confirmed high-quality RNA, the 'Procedure & Checklist – Iso-Seq Express Template Preparation for Sequel and Sequel II Systems' was followed (PN 101-763-800 version 02 [October 2019]). Two libraries were created from the same total RNA input (300 ng): one targeting 2 kb transcripts and one targeting >3 kb transcripts. Both libraries were SMRT sequenced using a Sequel binding plate 3.0, sequel sequencing plate 3.0 with a 20 hr movie time on a PacBio Sequel system on their own SMRT cell 1M v3 LR. The 2 kb library was loaded at 3 pM and generated 33.90 Gb and 409,187 ≥ Q20 Circular consensus reads (CCS). While the >3 kb library was loaded at 3 pM and produced 27.04 Gb and 248,304 ≥ Q20 CCS. All steps post RNA extraction were performed at the Next Generation Sequencing Platform, University of Bern. High-quality isoforms were generated from PacBio reads using the Iso-Seq analysis pipeline of single-molecule real-time (SMRT) Link (version 9), which generates circular consensus sequencing reads and then clusters and polishes the isoforms. Cd-hit-est version 4.6.8 (*Fu et al., 2012*) was used to remove redundancy due to different transcript isoforms. It was run using a sequence identity threshold of 95%. Afterward, the Coding GENome reconstruction Tool (cogent v 6.0.0, *Huo et al., 2020*) was used to further reduce redundancy and reconstruct the coding genome. In order to annotate the coding genome, the redundancy removed sequences were blasted against the nr database of NCBI using blastx version 2.9.0 (NCBI Resource Coordinators 2016). We aligned our reference non-redundant transcriptome to the benchmark universal single-copy orthologs database for Mollusca (BUSCOs) and obtained a score of 54.4% (49.8% single-copy, 4.6% duplicated, 1.9% fragmented). This indicated that the reference transcriptome has low redundancy (<5% duplicates) that is necessary for single-cell transcriptomic analysis but, on the other hand, is missing several genes.

## Cell dissociation for single-cell RNA sequencing

Embryos were first dissected out of their eggs in stages 28–30 (pre-hatchling stage) and their heads were isolated and put in nuclease-free phosphate buffer (PBS) on ice. The heads were then centrifuged for 5 min at 4°C, and the supernatant was carefully removed and replaced with a papain enzyme solution at a concentration 1 mg/ml in nuclease-free PBS. The tissue was then incubated for 30 min at room temperature (20–25°C) under continuous agitation. During the incubation period, the solution was pipetted up and down every 10 min to facilitate dissociation. The cell suspension was then centrifuged at 2000 rpm for 5 min at 4°C. The supernatant was removed and replaced by 1 ml of PBS containing 0.04% bovine serum albumin (BSA) to stop the enzymatic reaction. This last step was repeated one more time to ensure removal of the enzyme. The suspension was then filtered through a 40 µm Flowmi Cell strainer (Bel-Art H13680-0040). The filtered suspension was centrifuged at 2000 rpm for 5 min at 4°C. The supernatant was discarded, and cells were suspended in 100 µl of PBS 0.04% BSA with added RNAse inhibitor (0.1 µL/ml) and further dissociation was ensured by gently pipetting the entire volume approximately 200 times. Cell concentration and viability were assessed using a DeNovix CellDrop Automated Cell Counter with an Acridine Orange (AO)/propidium iodide (PI) assay.

## Single-cell capture and sequencing

Single-cell RNAseq libraries were prepared using a Chromium Single Cell 3' Library & Gel Bead Kit v2 or v3 (10X Genomics), according to the manufacturer's protocol (user guide). Two chips were loaded with the accurate volumes calculated based on the 'Cell Suspension Volume Calculator Table.' The initial single-cell suspension being estimated at >600 cells/µl, we targeted to recover a maximum 10,000 cells. Once GEMs were obtained, reverse transcription and cDNA amplification steps were performed. Sequencing was done on Illumina NovaSeq 6000 S2 flow cell generating paired-end reads. Different sequencing cycles were performed for the different reads, R1 and R2. R1 contained 10× barcodes and UMIs, in addition to an Illumina i7 index, and R2 contained the transcript-specific sequences. All steps were performed at the Next Generation Sequencing platform at the University of Bern.

## Single-cell RNAseq analysis

*10X Genomics* Cell Ranger version 3.0.2 (*Huo et al., 2020*) was used to map the single-cell RNAseq reads from 10X Genomics Chromium to the above assembled coding genome and count the number of reads overlapping with each gene to produce a count matrix for each sample. Afterward, the R package scater v. 1.14.0 (*McCarthy et al., 2017*) was used to automatically filter low-quality cells. Additionally, the gene NP-062835.1 was removed due to its abnormally high expression. The cells were filtered to keep only those with a gene per cell count comprised between 200 and 2000. The data was normalized based on a deconvolution method integrated into the R package scran v. 1.14.5 (*Lun et al., 2016*). Integration analysis, dimensionality reduction, and clustering were done using the R package Seurat v. 3.1.3 (*Stuart et al., 2019*). The data was normalized using the normalization method 'LogNormalize' with a scale factor of 10,000. The function 'FindVariableFeatures' was run using the selection method 'vst'. The data was then scaled, and we run principal component analysis function 'RunPCA'. Clustering was performed using the functions 'FindNeighbors' and 'FindClusters' using 25 dimensions and a resolution of 2. Upregulated genes for each cluster were identified using the function 'FindAllMarkers'. The 100 most differentially expressed markers are provided in *Supplementary file 1*. For the analysis of the nervous system, neurons were subclustered by selecting the cells of the clusters obtained previously that contained neuronal markers (Cl5, Cl7, Cl11, Cl13, Cl15, Cl20, Cl22, Cl27, Cl32). The cells were then reclustered with the same parameters as described above but using six dimensions and a resolution of 1.2. The 100 most differentially expressed markers in these neuronal subclusters are provided in *Supplementary file 2*. Subclustering of the cells used for the developmental trajectories was selected by subsetting the clusters identified as either neuronal, asc+, foxD+, or stem cells (Cl5, Cl7, Cl10, Cl11, Cl13, Cl15, Cl16, Cl17, Cl18, Cl19, Cl21, Cl26, Cl27, Cl32). In Seurat, the cells were reclustered using the same parameters as above but using six dimensions and a resolution of 1. The 100 most differentially expressed markers obtained with Seurat in these subclusters are provided in *Supplementary file 3*.

## Trajectory inference

Trajectory inference was used on the subset of cells that were reclustered from neurons, asc[+] cells, foxD[+] cells, or stem cells as described above. Monocle3 (version 1.2.9) was used to recluster the data and calculate trajectories. We used the R package 'Seurat_to_Monocle' to convert the Seurat object into a Monocle object (SeuratToMonocle3, version 0.1.0; https://github.com/Jackcava/Seurat_to_Monocle, *Jackcava, 2022*). The data was then preprocessed using the 'preprocess cds' function using PCA with 25 dimensions. Dimension reduction was performed using the 'reduce_dimension' function. Clustering was done using the function 'cluster_cells' with a resolution of 1e-3. Trajectories were calculated using the 'learn_graph' function and selecting Cl5 (see *Figure 3—figure supplement 4*) as the starting point. Trajectories were then plotted onto the UMAP using either cluster labels or pseudotime and the function "plot_cells" was used to show the expression of specific genes along the trajectory axis. The 100 most differentially expressed markers obtained using Monocle3 are provided in *Supplementary file 4*. Slingshot (*Street et al., 2018*, version 2.2.1) was used as an alternative to calculate trajectories. The 'slingshot' function was run with the embedded clustered Seurat object. Cl9 of the subclustered data (*Figure 3—figure supplement 5*) was selected as the starting cluster. The resulting trajectories were plotted onto the original clusters obtained with Seurat.

## Hybridization chain reaction

Probes for HCR ISH were ordered and designed by Molecular Instruments (*Choi et al., 2016*). Whole pre-hatchlings *L. vulgaris* animals (stages 28–30, *Arnold, 1965*) were dissected out of their eggs and were then anesthetized in a mix of 1.7% $MgCl_2$, 1% ethanol, and artificial seawater. After 20 min in the anesthetic solution, the animals were fixed in 4% formaldehyde for 48 hr at room temperature. Samples were then washed three times with phosphate buffer (PBS) and were dehydrated and stored in 100% ethanol at –20°C. Prior to HCR ISH experiments, samples were rehydrated in PBS in three 5 min steps. Our HCR ISH protocol used was adapted from the provided 'generic sample in solution' protocol (*Choi et al., 2016*) with the following modifications: a light proteinase K treatment was performed before pre-hybridization for 5 min at a concentration of 1 ul/ml in PBS at room temperature and subsequently washed three times with PBS. All volumes were reduced to 200 ul per well. Incubation times for hybridization and amplification were increased to 16–20 hr to increase signal (inappropriate for RNA quantification). DAPI was added together with one of the 5× SSCT washes. Double HCR ISH experiments were performed by combining B1-Alexa-488 hairpins with B2-Alexa-647 hairpins.

## Confocal imaging

Images were taken on Leica Stellaris 8 falcon or a Leica TCS Sp5 laser scanning confocal microscopes. All mosaic images were taken by automatically stitching 3 × 3 or 3 × 4 confocal image stacks with a 10% overlap. Laser power and gain were adjusted depending on the experiment to ensure optimal visualization. Images were treated with ImageJ where the contrast was adjusted. All images are optimized for qualitative assessment and are therefore not appropriately acquired or treated for quantitative analysis or comparisons.

## Acknowledgements

We wish to acknowledge the staff of the station Biologique de Roscoff, France, for organizing the collection and shipment of *L. vulgaris* embryos. We would like to thank the staff of the Next Generation Sequencing facility of the university of Bern for helpful inputs and their help and with the single-cell sequencing experiments.

## Additional information

### Funding

| Funder | Grant reference number | Author |
| --- | --- | --- |
| Schweizerischer Nationalfonds zur Förderung der Wissenschaftlichen Forschung | 310030_188471 | Simon G Sprecher |
| Schweizerischer Nationalfonds zur Förderung der Wissenschaftlichen Forschung | IZCOZ0_182957 | Simon G Sprecher |

The funders had no role in study design, data collection and interpretation, or the decision to submit the work for publication.

### Author contributions

Jules Duruz, Formal analysis, Validation, Investigation, Visualization, Methodology, Writing - original draft; Marta Sprecher, Formal analysis, Methodology; Jenifer C Kaldun, Al-Sayed Al-Soudy, Heidi EL Lischer, Geert van Geest, Investigation, Methodology; Pamela Nicholson, Rémy Bruggmann, Investigation, Methodology, Project administration; Simon G Sprecher, Conceptualization, Supervision, Funding acquisition, Writing - original draft, Project administration

## Author ORCIDs
Jules Duruz ⓘ http://orcid.org/0000-0003-1860-9546
Al-Sayed Al-Soudy ⓘ http://orcid.org/0000-0002-7830-9660
Heidi EL Lischer ⓘ http://orcid.org/0000-0002-9616-2092
Geert van Geest ⓘ http://orcid.org/0000-0002-1561-078X
Rémy Bruggmann ⓘ http://orcid.org/0000-0003-4733-7922
Simon G Sprecher ⓘ http://orcid.org/0000-0001-9060-3750

## Decision letter and Author response
Decision letter https://doi.org/10.7554/eLife.80670.sa1
Author response https://doi.org/10.7554/eLife.80670.sa2

---

## Additional files

### Supplementary files
• Supplementary file 1. Table of top 100 marker genes of all clusters. Top differentially expressed marker genes of all clusters.

• Supplementary file 2. Table of top 10 marker genes of neuronal clusters. Top differentially expressed marker genes of clusters with neuronal identity.

• Supplementary file 3. Table of top 100 marker genes of progenitors in Seurat. The 100 most differentially expressed markers obtained with Seurat in progenitor subclusters.

• Supplementary file 4. Table of top 100 marker genes of progenitors in Monocle. The 100 most differentially expressed markers obtained with Monocle in progenitor subclusters.

• MDAR checklist

### Data availability
Sequencing data have been deposited in GEO under accession code GSE200108.

The following dataset was generated:

| Author(s) | Year | Dataset title | Dataset URL | Database and Identifier |
|---|---|---|---|---|
| Duruz J, Sprecher M, Kaldun JC, Al-Soudy A, Tschanz-Lischer H, Nicholson P, Bruggmann R, Sprecher SG | 2022 | Molecular characterization of cell types in the squid Loligo vulgaris | https://www.ncbi.nlm.nih.gov/geo/query/acc.cgi?acc=GSE200108 | NCBI Gene Expression Omnibus, GSE200108 |

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
