## [Editor Report]

This article describes cell types in the head of the squid, *Loligo vulgaris*, through expression patterns of key genes identified in single-cell transcriptomics. This topic is generally of great comparative interest. The data presented here are convincing, and these valuable findings will contribute to a better understanding of the cephalopod nervous and sensory systems, providing a basis for future comparative and evolutionary research.

---

## [Decision Letter]

**Decision letter after peer review:**

Thank you for submitting your article "Molecular characterization of cell types in the squid *Loligo vulgaris*" for consideration by *eLife*. Your article has been reviewed by 3 peer reviewers, including Sonia Sen as the Reviewing Editor and Reviewer #1, and the evaluation has been overseen by a Reviewing Editor and Didier Stainier as the Senior Editor. The following individuals involved in the review of your submission have agreed to reveal their identity: Samuel Reiter (Reviewer #2); Astrid Deryckere (Reviewer #3).

Essential revisions:

While appreciating the relevance of this there were a set of common concerns raised by the reviewers that need to be addressed. These are:

1. The reference transcriptome: A high-quality reference transcriptome will be critical for any downstream analysis. Could the authors please provide data for how they validated their reference transcriptome? For example, the authors could report the BUSCO score for assessing the completeness of their assembly, a cumulative density plot of AED scores for annotation quality, and any other metric they choose. Could the authors also report what percent of their single cell reads mapped to the reference?

2. The scRNAseq data: Could the authors please provide data that would allow readers to assess the quality of their single-cell data? At a minimum, could they please provide violin plots depicting the range of UMIs, genes, and mitochondrial genes per cell?

3. Cluster-specific markers: Could the authors please provide dot plots for the top differentially expressed genes for all clusters? In addition to this, the authors have described a set of genes that they did not identify as cell-specific markers (for example, neurotransmission-related ones). Could they please show dot plots for these as well?

4. Trajectory Analysis: Could the authors please explain how they have chosen the cells used in this analysis, the clustering resolution they have used (please provide a dimplot), and which starting point they have selected? In addition to this, could they please analyze the differentially expressed genes along the trajectory to make their claim stronger? Since this data challenges the current state of the field, it would be useful to demonstrate the robustness of their results by verifying it by:

4a. Using different trajectory analysis methods (Monocle or FateID).

4b. Analysing more marker genes (in addition to foxD1) along the differentiation axis.

5. The writing: Could the authors please pay attention to some aspects of the text to help the reader place their work in the broader context of the field? Specifically:

a. There are missing references (see detailed reviews below), including three other recently posted cephalopod scRNAseq preprints. Could the authors please incorporate these references?

b. On occasion, the chronology of the text and figures don't match. There are also some figures that are currently not mentioned in the text. Could the authors please fix this?

c. Many of the methods are not sufficiently well explained. Could the authors please pay attention to this, particularly for bioinformatics? Could the authors motivate this better?

d. The Results section contains many points that would belong better in the Discussion section.

e. Conversely, the discussion as it currently stands, is a summary of the results. Could the authors please revisit their discussion to place their work and findings in the context of the growing field of cephalopod neurobiology?

*Review #1 (Recommendations for the authors):*

1. I was concerned about the quality of the scRNAseq data for a few reasons:

- The number of genes per cells (200-2000) seems low to me.

- The clusters don't really resolve.

- Many genes that one would have expected to pick up, for example, neurotransmitter-related genes for the neuronal cluster, were not picked up.

So, could the authors please provide the violin plots depicting the range of UMIs, genes, and mitochondrial genes per cell?

2. It was not clear to me how the authors picked the marker genes. Could they provide the top 20 DEGs for each of the clusters?

3. The signal in each of the images could be enhanced. To orient the reader, it would also be useful to have an inset of each of the expression patterns in the whole head. Some of these are really nicely represented in the supplementary information.

4. The authors should refer to the three recent preprints that have used scRNAseq in other cephalopods and, in the discussion, place their work in this context.

5. In general, could the authors elaborate their method sections? Many fairly important aspects are too briefly described – for example, the reference transcriptome and its validation and the selection of marker genes, among others.

*Reviewer #2 (Recommendations for the authors):*

The core of this study is the analysis of scRNAseq data. Here the authors assembled a transcriptome de novo. I wonder about the quality of this assembly, as it will affect all downstream analysis. The other 3 studies (Styfhals et al. bioRxiv 2022.01.24.477459; doi: https://doi.org/10.1101/2022.01.24.477459; Songco-Casey et al. bioRxiv 2022.06.11.495763; doi: https://doi.org/10.1101/2022.06.11.495763; Gavriouchkina et al., bioRxiv 2022.05.26.490366; doi: https://doi.org/10.1101/2022.05.26.490366) all refer to the necessity of a high-quality reference genome for a high-quality transcriptome. If this is not the case here, it would be useful to quantify this.

I found the analysis of stem cells and progenitors quite interesting. The authors profile specific markers for cell types, map these cell types onto interesting anatomical regions, and analyze developmental trajectories defined transcriptomically.

In other parts of the paper, I was surprised that many cell types could not be defined based on certain marker genes/small combinations of genes. Figure 2b, for example, shows that clusters have a great deal of overlapping expression among 'marker' genes. This would deviate quite a bit from the other scRNAseq studies I am familiar with. I could see a few possible explanations: (1) This is a product of low-quality transcriptome assembly/single-cell data (2) This is a product of the immature developmental stage of the animal, and (3) This reflects a cephalopod-specific transcriptomic signature.

I doubt possibility 3, as the other 3 cephalopod scRNAseq papers describe cell types well differentiated (eg. By neurotransmitter expression, Figure 2 in Songco-Casey et al. 2022, Figure 3 in Gavriouchkina et al. 2022). The fact that specific neurotransmitters are often not well localized to specific cell types here calls for an explanation. Similarly, the lack of differentiation of neural types even when analyzed separately from other cell types (Figure 2c). Many phototransduction genes are expressed in Cluster 32, and these genes are expressed, specifically in the retina. They could potentially be used as a sanity test for the scRNAseq: Are these genes only found in cluster 32?

The lack of specific marker genes makes some of the FISH experiments in this manuscript less informative than others. Many plots show anatomical localization of a gene broadly expressed in many cell types, meaning we cannot conclude how transcriptomically identified clusters map spatially. In other cases, FISH is used to mark individual cell types. In some cases, it is not clear which situation we are looking at. It would be useful to have a quantification of how specifically the marker genes used in FISH experiments mark individual clusters.

The authors observe a similar anatomical expression of fmrf and reflectin-8 in the olfactory organ. Because these have been linked to chromatophore and iridiophore control, they argue that this suggests that chromatophores and iridiophores are modulated through the olfactory organs. I find this plausible but the logic questionable. FMRF has many functions unrelated to chromatophore modulation, so just observing its presence seems insufficient to make statements about the neural control of chromatophores. Reflectin-8 is shown later in the manuscript to be broadly expressed in the epidermis and brain (Figure 5c).

I was not clear why the expression of elav-like 1 in the epidermal lines 'informs on their potential mechanosensory function'. With elav-like 1 expressed very broadly across cell types and anatomical locations.

The expression of the cilia-associated protein, rab3 and synaptogyrin-1, and other cilia/flagella markers in cluster 27 suggested that this corresponds to a mechanosensory population. To test whether this is neural, the authors take the subset of cells in the dataset expressing amyloid β- binding-like (a neural marker) and see whether a subset of these also expresses the collection of sensory markers. This seems round about to me. Why not look directly at whether the cells in cluster 27 express amyloid β- binding-like? If the probability of co-expression is high, then concluding that these cells are neurons seems warranted. If the probability is lower, then the conclusion does not seem warranted. The author's existing analysis does not address the possibility that non-neurons (those not expressing amyloid β- binding-like) express the collection of sensory markers.

Line 63: 'Coleoid cephalopods, which have internalized or lost their shell over evolutionary time 64 were able to swim more easily and could exploit other niches.'

Not clear what 'other' refers to. The niches of ammonoids? Not clear whether it was the loss of shell that led to niche exploitation or the other way.

Line 93 "… as a first step to understanding cephalopod evolution". I'm not clear why a molecular comparison would be a first step to understanding cephalopod evolution. What then is the non-molecular evolutionary story given earlier in the introduction?

Figure 2 Are the Umap plots in 2c re-estimated on the subset of data in A? If so, I don't see a description of this in the methods section, and I am surprised at the lack of cell type differentiation among neurons.

Figure 4D is described as 'feature plots'. Is this UMAP? How are the plots on the right-hand side generated?

The developmental trajectory analysis using slingshot could be better explained in text/methods/figure legend and not easily interpretable from Figure 3e if one is not familiar with the technique

'Additionally, the cephalopod-specific chromatophores, which are colored or iridescent, have been suggested to be modulated by the nervous system'. Maybe I am misunderstanding the sentence, but as I read it this is an understatement of what we know. Over 50 years of work have profiled the neural control of chromatophores in detail.

Some of the supplementary figures are quite dark. Why not increase the brightness of the images? Others like Figure 4 supplement 1 are beautiful.

*Reviewer #3 (Recommendations for the authors):*

First of all, I would like to congratulate the authors on the extensive expression analysis study performed in this manuscript. However, I believe that this dataset offers so many more opportunities to better understand cell types in cephalopods, and in general, the data could be analysed and interpreted in more detail.

Since the field of scRNAseq is moving incredibly fast, the authors should be careful with claims of being first (e.g. Abstract line 34). Preprints of scRNAseq data in the nervous system of other cephalopods have been recently published (Styfhals 2022, Gavriouchkina 2022, Songco-Casey 2022).

The manuscript is missing key citations to support the results and to provide a comparative view of gene expression in other cephalopods. For example, the expression patterns of Elav, SoxB1, Asc have been reported during the development of octopus, squid, and cuttlefish, and could enhance the interpretation of the results observed in Loligo.

In general, the reader needs to be proficient in the brain areas of cephalopods in order to follow the results. Therefore, the manuscript would benefit from a short introduction of the main brain lobes in cephalopods to be accessible to a wider audience, either in the introduction or in the results in Figure 1C.

Please make sure to refer to the figures and figure panels chronologically. Multiple figure panels are not referred to throughout the text (e.g. the first figure described is Figure 1B and there is no notion to Figure 1A, similar to Figure 2, …).

The authors distinguish 33 cell clusters when using 25 pca's and a resolution of 2. The UMAP, however, shows most cells in one single, central blob. This is striking knowing how different connective tissue is from neurons. Did the authors try other methods for clustering and dimensionality reduction, or change the parameters in the current algorithm to try to resolve the data better? It would also be helpful to specify the filtering parameters for filtering out low-quality cells in the methods section.

In the first section of the results, the authors list a handful of enriched genes used to differentiate the major cell type categories. However, little explanation is given to why these genes were chosen and a comprehensive overview supporting the claims is missing. It would help the reader if they would add a plot (e.g. a DotPlot) to support these data. In addition, a UMAP plot with cells colored according to cell cycle could clearly show where the progenitor cells are located on the UMAP.

To avoid going back and forth between cell type categories in the first results paragraph, it would be helpful to describe Cl 27 (lines 202 onwards) and Cl 32 (lines 207 onwards) with the other nervous system cells, and Cl 29 (lines 213 onwards) with the epidermis section.

In order to interpret HCR expression patterns, the authors should add axis labels to the figures. In addition, I understand that the figure panels are small to add annotation of the brain regions, but since we are looking at different levels through the brain, either a schematic with the figure or annotation would really help.

The authors should explain the analysis performed to obtain the UMAP(?) plots in Figure 2C. Which cells were selected? Were they re-clustered? What do the clusters look like?

Did the authors perform image manipulation e.g. Figure 2C and 2D for the DAPI channel? Why is DAPI absent from the neurons in the tetraspanin-8 HCR, or the serotonin transporter staining?

In Figure 3B, it would be helpful to add some genes involved in proliferation, as exemplified in the text.

The authors claim that neural stem cells undergo neuronal differentiation and then migrate to the nervous system. However, evidence is missing as the authors do not show where progenitor cells stop dividing and start to differentiate.

The use of the marker foxD1 for differentiation neurons is not well supported. Additional marker analysis is required before this can be claimed.

The authors need to explain which clusters were used for trajectory analysis using Slingshot and how the data were reclustered. Were non-neuronal stem cells included? This might skew the analysis. Importantly, the method was not described in the methods section. It is critical for the readers to know which endpoints were selected in the slingshot algorithm. Off note, the slingshot is very dependent on clustering resolution and will try to connect all clusters. Have the authors tried a different pseudotime analysis method such as Monocle or FateID? Additionally, can the authors elaborate on which transcription factors are steering this differentiation? In the current state, the analysis is preliminary and contradicts present data on the origin and trajectory of neurons in the cephalopod brain (see comment below; Koenig et al. 2016, Deryckere et al. 2021).

With the current interpretation of the data, the authors claim that foxd1 cells are an intermediate developmental state of neurons. This would be very striking since the cortex of the optic lobe has been described as being the inner retina. Additionally, it is contradicting the expression of amyloid β-binding like in the optic lobe cortex, which the authors say is only expressed in differentiated neurons. Is amyloid β-binding-like expressed in clusters 15, 19, and 20? How about other markers for differentiated neurons? What would all those intermediate cells do there? Do the authors think they migrate into the medulla or even the central brain? This is contradicting existing literature on the origin of neurons in the cephalopod brain (Koenig 2016, Deryckere 2021) and requires in-depth discussion.

The authors prepared beautiful summary diagrams in Figures 3 and 4. However, they should be referred to in the main text.

After performing extensive HCR expression studies, can the authors go back to their scRNAseq data and better annotate the different clusters?

In order to better understand neuronal diversity and neural cell types (as the authors claim), the dataset has to be analyzed in more depth. 33 clusters/cell types seem low given the number of cells that were sequenced and the different tissues that were sampled. Can the authors identify the different brain regions when subclustering the neurons? Can they identify and annotate different neural cell types?

The authors present several striking findings. However, at the moment, the Discussion section explains the results obtained in the paper, but does not place the acquired data within the current state-of-the-art. Parts of the results should be moved to the discussion and the discussion should be elaborated on.

Specific comments:

Line 76: "by the use its specialized iridescent cells" misses an of

Line 97: it would be helpful for the readers to add the "pre-hatchling stage" to stage 28, as was done in the Results section for the audience to have a better idea of what stage was studied.

Throughout the text (starting line 219) please write "HCR ISH" instead of just "HCR". The chain reaction method is not specific to in situ hybridization and can also be used in immunohistochemistry and others.

Line 101: "in all animals" please be more specific; not all animals have neurons.

Line 110: "marking distinct stages of neural development", neural differentiation would be more appropriate.

Line 126: Add a reference to Figure 1A to show what the head region is.

Line 133 says 34 cell clusters were identified, but Figure 1B shows 33.

Line 145: "smooth muscle-like cells (Cl26)" should be Cl25.

Line 156-159: add a reference for this statement, maybe a specific example.

Line 167-171: "a gene whose function is homologous to …" and " exact function … is not known". This is contradictory. Do you mean that the gene's sequence is homologous?

Line 199-201: with the comments below on SoxB1 and Ascl1, this sentence needs to be revisited.

Line 239-240 and also further in the text: "in the medial portion of the optic lobes" do the authors mean the medulla (in contrast to the cortex (inner and outer granular layer))? Medial portion is not clearly defined and the cortex also stretches quite medially. (also line 278, 540).

Lines 263 and 271: please rearrange the supplementary figure so you can refer to panels A-D first. In general, please refer to figures chronologically.

Line 298 with Figure 2D: please switch panels of LIM and serotonin, so they follow the text.

Line 326: this part covers more than stem cells and progenitors (line 375 and onwards).

Line 327: please add an introductory sentence to this new paragraph; rephrase the first sentence (grammatically incorrect).

Line 338-340: this is an overstatement. What identity are the authors referring to? Genes involved in proliferation etc do not give identity to stem cells and Cl02 and Cl14 are also actively cycling. Do the authors suggest only Cl10 and Cl17 are stem cells? Can the authors show more evidence?

Lines 350-359: I would suggest moving this part to the discussion and incorporating the following comments:

Lines 347-352: The expression of Asc is not as well conserved as the authors suggest. Indeed, it has some proneuronal role, but the level of differentiation in the cell it is expressed is very different across the animal tree. In *Drosophila* as in other non-insect arthropods, Asc is expressed in quiescent ectodermal cells to drive them into the neural lineage (formation of a neural progenitor cell) (e.g. Cabrera et al., 1987; Skeath and Carroll, 1992). In contrast, in vertebrates and also in spiralia such as capitella teleta, Ascl1 is expressed in neural progenitor cells to drive them into differentiation (e.g. Bertrand 2002; Sur 2020). Ascl1 expression has been studied in octopus and seems to have a similar function as in capitella, not *Drosophila* (Deryckere 2021). This scRNAseq dataset has the power to test this hypothesis and figure out whether, in cephalopods, as is expressed in proliferating and not quiescent cells.

Lines 354-355: this conclusion does not reflect the data shown. The authors are showing proliferating stem cells, not the location of differentiation. In octopus, differentiation seems to happen in the transition zones (NeuroD cells, Deryckere 2021). It would be interesting to locate those cells in your dataset.

Lines 356-359: Again here, one needs to be careful to extrapolate the functionality of a gene in one species to other species. In invertebrates, SoxB1 is also expressed in differentiated neurons (e.g. Le Gouar et al., 2004; Semmler et al., 2010). Additionally, expression of SoxB1 has been described in Sepia (Focareta and Cole, 2016) and octopus (Deryckere 2021), indicating expression in neurons (with some interesting discrepancies). Furthermore, be careful with the use of the word "neuroblast" because it refers to different types of cells in different animals.

Line 361: "Interestingly" it seems a general theme for cephalopods, and maybe invertebrates in general (see comment above).

Line 367: Data are missing to show where the neural progenitor cells differentiate.

Line 370-374: in line with this, SoxB1 expression is also found in sensory epithelia of sepia and octopus (e.g. suckers and statocyst; Baratte, S. and Bonnaud 2009; Deryckere 2021).

Line 376-378: add a reference and elaborate. Regulators of development is a very general process. In addition "for this reason, these cells were presumed to be differentiating cells": this is an overstatement and needs additional gene expression patterns to support this presumption.

Line 382-385: optic lobe cortex = inner and outer granular layers, please adjust and rephrase. From Figure 3C upper panel, it seems that Foxg1 is also expressed in the medulla. Or is this all cortex? Annotation would be helpful here.

Line 386: this is an overstatement. Most neurons in the brain express elav1 and the function of foxD1 is very broad. Additional markers (like for example NeuroD) are required to support this claim.

Lines 394-400: Figure 3D is not mentioned in the text and seems substantial. The authors might also want to revise it with the comments above and in public review.

Lines 408-411: please add references to support these statements.

Lines 419-421: Why did the authors choose to map the expression of Pax9? Koenig et al. studied the eye in Doryteuthis and mapped the expression of several genes important for eye development. A reference to this work would be appropriate here. Pax9 is not an eye gene in vertebrates and invertebrates (in contrast to Pax2/6). In which clusters is it expressed (again here, a dotplot of the genes under study would be helpful)? The level of the image in Figure 4A for Pax9 is also different than the others, why is this? In which cells of the eye is Pax9 expressed?

Line 448: Figure 5B should be Figure 4B I think.

Line 457: can the authors verify that they selected cells with expression level >3? From the legend in Figure 4D, the expression level of 3 seems present in a lot of cells (it looks more like the selected level was 5). In addition, the authors should explain how they obtained the graphs on the right. It looks like they underwent an extra transformation? Or are they just distorted?

Line 459, and together with the comment above: It would be helpful to be able to check the cluster numbers. For example, are cilia-associated protein cells from Cl27? A dotplot or heatmap might be more convincing to support these statements.

In addition, please add a reference to Figure 4D here.

Line 468-470: egf-like does not seem to be expressed in Cl03 and 31 from Figure 5A as suggested in the text.

Lines 471-493: please add the discussed genes to the dotplot in Figure 5A or supplement.

Line 541: It seems that cells in Cl27 do express neuronal markers. Please revise.

Line 545 and onwards: please not only discuss the results obtained but place the discussion in the state-of-the-art literature. Several statements have already been proposed in other papers which deserve to be referenced (e.g. lines 559, 568). Multiple genes mapped with HCR in this manuscript have been described in other cephalopods and assigned a preliminary function (e.g. Buresi 2013, 2014; Shigeno 2015; Koenig 2016; Focareta 2016; Deryckere 2021 and others). Please acknowledge their presence in appropriate sections, and, if the authors which, they might add a paragraph describing consistency or differences in expression.

Line 562: in line with previous comments in the results, please revise. This is based on the functionality of asc in *Drosophila*, but not other animals. Additionally, in the opinion of the authors, what are the asc+ e2f3- cells?

Line 713: please add the parameters used to filter low-quality cells.

Line 722: please add additional information on how cells were subclustered (and re-clustered?). Please add a description of the slingshot analysis.

Figures:

In general, please rearrange so each panel comes chronologically in the Results section. Make sure to mention each panel in the text.

Add axes to the confocal images and consider annotation.

Figure 1:

A) Is this the ventral view instead of the dorsal view? The funnel is visible?

Please arrange the abbreviations alphabetically. Ibl and sbl are missing.

Figure 2:

Line 790: UMAP with clusters.

C) Add legend to the 1→6 expression? Bar. It is not clear how these featureplots were calculated. Which clusters were selected? Were they re-calculated? Please show the dimplot with cluster annotation. What is on the x- and y-axes? UMAP? t-SNE?

D) Please rephrase 'that were not represented in the scRNAseq data': do the authors mean not differentially expressed? Were the genes absent from the reference transcriptome?

Figure 3:

A) Why are Cl2 and Cl14 encircled?

C) Explain what the red encircled areas are.

Figure 4:

D) Double check that 3 is the threshold. The curly bracket is confusing (it reads like the right panels feed into the left). What are the x- and y-axes? Similar comment to Figure 2C.

---

## [Author Response]

Essential revisions:While appreciating the relevance of this there were a set of common concerns raised by the reviewers that need to be addressed. These are:1. The reference transcriptome: A high-quality reference transcriptome will be critical for any downstream analysis. Could the authors please provide data for how they validated their reference transcriptome? For example, the authors could report the BUSCO score for assessing the completeness of their assembly, a cumulative density plot of AED scores for annotation quality, and any other metric they choose. Could the authors also report what percent of their single cell reads mapped to the reference?

We thank the reviewers for pointing this out. We have now included metrics about the reference transcriptome that was used for this study. We discuss the implications of using a transcriptome instead of a genome for the downstream analysis and emphasize this point as the main limitation of our analysis. We now include the suggested parameters and report the BUSCO score (s 876-882) and the mapping rate of the reads to the reference in figure 1 —figure supplement 1 and discuss that the low mapping rate is directly linked to the coverage of the reference transcriptome. We agree that a high-quality genome would be better for single cell RNAseq analysis, however transcriptomes provide an alternative if a genome is not available. We are indeed planning in the future to sequencing the *Loligo vulgaris* genome, given the genome size of many cephalopods this remains still likely a major effort.

2. The scRNAseq data: Could the authors please provide data that would allow readers to assess the quality of their single-cell data? At a minimum, could they please provide violin plots depicting the range of UMIs, genes, and mitochondrial genes per cell?

We now added a figure supplement (figure 1 —figure supplement 1) that shows violin plots for the gene-per-cell counts, number of UMIs and percentage of mitochondrial genes to better assess the quality and variability of the data.

3. Cluster-specific markers: Could the authors please provide dot plots for the top differentially expressed genes for all clusters? In addition to this, the authors have described a set of genes that they did not identify as cell-specific markers (for example, neurotransmission-related ones). Could they please show dot plots for these as well?

We added a supplementary figure with large dot plot that shows the top ten differentially expressed genes in each of the 34 clusters to help the reader identify the genes discussed in this study as well as potential other markers that are not discussed (figure 1 —figure supplement 2). We also provide dot plots for the neuronal markers that were not cluster-specific but for which we performed HCR ISH experiments. These plots are part of a new supplement (figure 2 —figure supplement 8).

4. Trajectory Analysis: Could the authors please explain how they have chosen the cells used in this analysis, the clustering resolution they have used (please provide a dimplot), and which starting point they have selected? In addition to this, could they please analyze the differentially expressed genes along the trajectory to make their claim stronger? Since this data challenges the current state of the field, it would be useful to demonstrate the robustness of their results by verifying it by:4a. Using different trajectory analysis methods (Monocle or FateID).

This is indeed an important point. We agree with the reviewers that the explanation for the trajectories was not precise enough. We now provide more detailed information about how the trajectories were calculated and which parameters were used in a new section of the material and methods section titles "trajectory inference" (lines 961-981). We replicated the results obtained with slingshot by running the analysis using monocle3. We decided to include the monocle-based trajectories in the main figure (Figure 3) and the results obtained are described in the manuscript (lines 461-489).

We also identified additional genes that can be followed along the trajectory axes (Figure 3 —figure supplement 4). The analysis using slingshot is now part of the supplementary data and we could also follow the expression of several genes along the trajectory on a dot plot (Figure 3 —figure supplement 5, lines 490-501).

5. The writing: Could the authors please pay attention to some aspects of the text to help the reader place their work in the broader context of the field? Specifically:a. There are missing references (see detailed reviews below), including three other recently posted cephalopod scRNAseq preprints. Could the authors please incorporate these references?b. On occasion, the chronology of the text and figures don't match. There are also some figures that are currently not mentioned in the text. Could the authors please fix this?c. Many of the methods are not sufficiently well explained. Could the authors please pay attention to this, particularly for bioinformatics? Could the authors motivate this better?d. The Results section contains many points that would belong better in the Discussion section.e. Conversely, the discussion as it currently stands, is a summary of the results. Could the authors please revisit their discussion to place their work and findings in the context of the growing field of cephalopod neurobiology?

We thank the reviewers for their suggestions for improving the writing. We added the missing references for the three recent cephalopod single-cell studies. We also added additional references to better support the rationale and the interpretation of the data. We checked the chronology of the figures and made sure that all figures and supplements are cited in the text. We developed the methods section to include more detail about the downstream analysis of the data and the bioinformatics. As stated above, we now also give more detailed information about the calculation of trajectories for figure 3 (Materials and methods, trajectory inference). We also included the information that was previously missing about the sub-clustering parameters used for figures 2 and 3 in the Materials and methods on lines 944-957. We developed the discussion to better place our findings within the existing literature and to further address possible issues with the analysis. Because the results are very descriptive by nature, we decided to keep some of the discussion-like elements in the Results section to provide the reader with a minimum of context before reaching the discussion. However, all the points mentioned in the results are then discussed more in depth in the discussion.

Review #1 (Recommendations for the authors):1. I was concerned about the quality of the scRNAseq data for a few reasons:- The number of genes per cells (200-2000) seems low to me.- The clusters don't really resolve.- Many genes that one would have expected to pick up, for example, neurotransmitter-related genes for the neuronal cluster, were not picked up.So, could the authors please provide the violin plots depicting the range of UMIs, genes, and mitochondrial genes per cell?

Thank you for pointing this out. As discussed in point 1 of the joint reviews, we now provide the metrics for the quality of the RNAseq experiments as well as the reference transcriptome and the details concerning mapping rate of the single-cell reads (Figure 1 —figure supplement 1, lines 132-153). The quality of the metrics of the single cell RNAseq experiments are of high standards. Based on combined metrics of transcriptome and single-cell RNseq, we propose that the lack of some relevant marker genes and neuron-specific markers is caused by coverage of the reference transcriptome. This is likely due to low expression level and transcript abundance of some critical markers, indeed property that differs between transcritpomes and genomes. We now also discuss that the use of a transcriptome versus genome impacts the analysis.

2. It was not clear to me how the authors picked the marker genes. Could they provide the top 20 DEGs for each of the clusters?

Markers were chosen typically from the top differential expressed genes. We now provide a list of the top hundred DEGs for each cluster in supplementary table 1. Additionally, the expression of the top 10 differentially expressed genes for each cluster can now be seen on a dot plot included as figure 1 —figure supplement 2.

3. The signal in each of the images could be enhanced. To orient the reader, it would also be useful to have an inset of each of the expression patterns in the whole head. Some of these are really nicely represented in the supplementary information.

Thank you for raising this point. The contrast was enhanced on all the confocal micrographs. To help the reader understand the pictures shown on the main figures, we provide an inset showing the location of the micrograph in the animal as well as anatomical annotations, similarly to the figure supplements. We additionally include a description of the anatomy of the *Loligo* nervous system in the Results section on lines 253-266 with a supporting figure (figure 2 —figure supplement 1) to help orient the reader. When available, the overviews of the tiled confocal micrograph showing the whole head are available in the figure supplements.

4. The authors should refer to the three recent preprints that have used scRNAseq in other cephalopods and, in the discussion, place their work in this context.

We now cited these three papers and have placed out analysis in the context of these studies (Lines 98, 656 and 679).

5. In general, could the authors elaborate their method sections? Many fairly important aspects are too briefly described – for example, the reference transcriptome and its validation and the selection of marker genes, among others.

We indeed agree with this point. We have further and extended developed the results and Materials and methods section to include more details about the transcriptome quality assessment as well as the mapping rate of the reads to the transcriptome (Results: lines 132-153). We added a more detailed description of the clustering and sub-clustering process done in Seurat and we wrote a new section to describe the way the trajectories from figure 3 were obtained (Materials and methods: lines 961-981).

Reviewer #2 (Recommendations for the authors):The core of this study is the analysis of scRNAseq data. Here the authors assembled a transcriptome de novo. I wonder about the quality of this assembly, as it will affect all downstream analysis. The other 3 studies (Styfhals et al. bioRxiv 2022.01.24.477459; doi: https://doi.org/10.1101/2022.01.24.477459; Songco-Casey et al. bioRxiv 2022.06.11.495763; doi: https://doi.org/10.1101/2022.06.11.495763; Gavriouchkina et al., bioRxiv 2022.05.26.490366; doi: https://doi.org/10.1101/2022.05.26.490366) all refer to the necessity of a high-quality reference genome for a high-quality transcriptome. If this is not the case here, it would be useful to quantify this.I found the analysis of stem cells and progenitors quite interesting. The authors profile specific markers for cell types, map these cell types onto interesting anatomical regions, and analyze developmental trajectories defined transcriptomically.In other parts of the paper, I was surprised that many cell types could not be defined based on certain marker genes/small combinations of genes. Figure 2b, for example, shows that clusters have a great deal of overlapping expression among 'marker' genes. This would deviate quite a bit from the other scRNAseq studies I am familiar with. I could see a few possible explanations: (1) This is a product of low-quality transcriptome assembly/single-cell data (2) This is a product of the immature developmental stage of the animal, and (3) This reflects a cephalopod-specific transcriptomic signature.I doubt possibility 3, as the other 3 cephalopod scRNAseq papers describe cell types well differentiated (eg. By neurotransmitter expression, Figure 2 in Songco-Casey et al. 2022, Figure 3 in Gavriouchkina et al. 2022). The fact that specific neurotransmitters are often not well localized to specific cell types here calls for an explanation. Similarly, the lack of differentiation of neural types even when analyzed separately from other cell types (Figure 2c). Many phototransduction genes are expressed in Cluster 32, and these genes are expressed, specifically in the retina. They could potentially be used as a sanity test for the scRNAseq: Are these genes only found in cluster 32?

We thank the reviewer for these valuable comments. In light of the documented impact of reference quality on the downstream analysis, we now provide additional information regarding the quality of the single-cell reads and the reference transcriptome (Results: lines 132-153, figure 1 —figure supplement 1). As mentioned above (joint reviewer comments) we indeed fully agree that a reference genome would be beneficial for the downstream analysis of sc-RNAseq data. We aim to take this on the task of generating a *Loligo vulgaris* genome in the future. With the transcriptome as reference, we cannot exclude that some cell types do not resolve because of the developmental stage of the animal. The analysis of the completeness of the transcriptome as measured by its BUSCO score, it seems a likely explanation for the lack of cluster resolution. This is probably most relevant for the absence of some genes such as makers of neurotransmitter synthesis pathway or patterning transcription factors. However, we feel that the current analysis provides a substantial advance, particularly in an otherwise molecularly underexplored taxon. With the reference markers we can identify and characterizes most clusters. Most importantly we feel that the histological correlation and mapping of cell types in situ is powerful. When looking at the example of phototransduction genes, they are very specifically expressed in cluster 32 but they can also be found in other cell types at much lower expression levels. The cell-type specificity observed with HCR ISH in the retina may not be the full picture. As in most other species one might expect these genes to be also present in other cells and other tissues, as exemplified in recent years in model organisms such as *Drosophila* or mouse.

The lack of specific marker genes makes some of the FISH experiments in this manuscript less informative than others. Many plots show anatomical localization of a gene broadly expressed in many cell types, meaning we cannot conclude how transcriptomically identified clusters map spatially. In other cases, FISH is used to mark individual cell types. In some cases, it is not clear which situation we are looking at. It would be useful to have a quantification of how specifically the marker genes used in FISH experiments mark individual clusters.

Thank you for raising this important point. We now provide additional information regarding the cell type specificity of markers of interest in the supplementary table 1 and in figure 1 —figure supplement 2. Generally, the choice of marker genes was more based on the availability of information and literature regarding their function or cell type specificity rather than a purely mathematical selection. This explains why some of the genes picked in our analysis are not always the markers with the lowest p-value but are the ones that have been previously associated with cell types of interest, development and, in certain cases, that are candidates for marking novel cell types as exemplified by the sections about *papilin* and *madnf* expression. It is indeed an intriguing idea to start probing for instance cephalopod or *Loligo* specific genes in the future.

The authors observe a similar anatomical expression of fmrf and reflectin-8 in the olfactory organ. Because these have been linked to chromatophore and iridiophore control, they argue that this suggests that chromatophores and iridiophores are modulated through the olfactory organs. I find this plausible but the logic questionable. FMRF has many functions unrelated to chromatophore modulation, so just observing its presence seems insufficient to make statements about the neural control of chromatophores. Reflectin-8 is shown later in the manuscript to be broadly expressed in the epidermis and brain (Figure 5c).

This is indeed important and we fully agree with your point. We apologize for this overstatement. We have now reformulated this sentence. It now reads: " FMRFamide has been previously shown to be involved in the control of chromatophores in squids (Loi and Tublitz 2000; Sweedler et al. 2000; Satpute-Krishnan et al. 2006; Mobley et al. 2008)".

I was not clear why the expression of elav-like 1 in the epidermal lines 'informs on their potential mechanosensory function'. With elav-like 1 expressed very broadly across cell types and anatomical locations.

Several genes that we analyzed are highly expressed in some clusters but are also expressed at lower levels in other tissues. As discussed above with the example of phototransduction genes, HCR experiments tend to only reveal cell-types in which the gene is highly expressed. For this reason, we interpreted the high signal from the epidermal lines as an indication of the presence of neurons in this structure. The mechanosensory modality of these neurons is based on studies done in other cephalopods. We reformulated the sentence to make this clearer.

It now reads: " The expression of *elav-like 1* in this structure in *L. vulgaris* is consistent with their proposed mechanosensory function other cephalopods."

The differential expression of the two elav genes is quite interesting and might reflect different functions as recent results in *Drosophila* of the three Elav-family members (Elav, Fne and Rbp9) suggest (Lee et al., 2021,Plos Genetics).

The expression of the cilia-associated protein, rab3 and synaptogyrin-1, and other cilia/flagella markers in cluster 27 suggested that this corresponds to a mechanosensory population. To test whether this is neural, the authors take the subset of cells in the dataset expressing amyloid β- binding-like (a neural marker) and see whether a subset of these also expresses the collection of sensory markers. This seems round about to me. Why not look directly at whether the cells in cluster 27 express amyloid β- binding-like? If the probability of co-expression is high, then concluding that these cells are neurons seems warranted. If the probability is lower, then the conclusion does not seem warranted. The author's existing analysis does not address the possibility that non-neurons (those not expressing amyloid β- binding-like) express the collection of sensory markers.

We thank the reviewer for pointing out this issue. We indeed realized that this is a circular argument in this case as cells were originally identified as neurons on the basis of the expression of markers such as amyloid-β binding. The intent was to visually represent the co-expression of these newly identified sensory markers with neuronal markers. We changed figure 4 to plot the expression of the sensory genes in the set of neuronal genes used in figure 2. We reformulated the sentence to highlight that this is not a proof of their neuronal identify but rather a demonstration that the hypothesis regarding sensory cell types is consistent with the hypotheses made in the previous sections. It now reads: " The results highlight the co-expression of neuronal markers and sensory markers, therefore supporting their sensory identity (figure 4D)." (lines 550-553).

Line 63: 'Coleoid cephalopods, which have internalized or lost their shell over evolutionary time 64 were able to swim more easily and could exploit other niches.'Not clear what 'other' refers to. The niches of ammonoids? Not clear whether it was the loss of shell that led to niche exploitation or the other way.

We now specify that coleoid cephalopods were able to access resources in open waters, which was not the case for shelled cephalopods. It is not possible to confidently state that one event caused the other in this case. However, the exploitation of different niches, which was only possible because of the increased mobility of coleoid cephalopods, favored the differences in nervous system anatomy that can be observed today between Nautilus and coleoid cephalopods.

Line 93 "… as a first step to understanding cephalopod evolution". I'm not clear why a molecular comparison would be a first step to understanding cephalopod evolution. What then is the non-molecular evolutionary story given earlier in the introduction?

We agree that this statement is not accurate. We reformulated this sentence to state instead: "These comparisons constitute a major step for understanding the molecular basis of cephalopod evolution" (lines 92-93).

Figure 2 Are the Umap plots in 2c re-estimated on the subset of data in A? If so, I don't see a description of this in the methods section, and I am surprised at the lack of cell type differentiation among neurons.

The clusters of figure 2A were indeed re-clustered to obtain the plots of figure 2C. We have now included a description of this process and the parameters used for subclustering in the material and methods (lines 944-950). Additionally, the UMAP with the new clusters is available in figure 2 —figure supplement 8.

Figure 4D is described as 'feature plots'. Is this UMAP? How are the plots on the right-hand side generated?

We modified figure 4D and the plots shown there are UMAPs representing gene expression in the subset of cells as for figure 2 (figure 2 —figure supplement 8, Materials and methods: lines 944-948).

The developmental trajectory analysis using slingshot could be better explained in text/methods/figure legend and not easily interpretable from Figure 3e if one is not familiar with the technique

We have included a section in the material and methods section to explain how these results were obtained (Materials and methods: lines 961-981). In addition, results were replicated using a different trajectory inference method monocle (Results: lines 461489, Materials and methods: lines 961-976). The monocle data is now part of the main figure (figure 3) and the slingshot data obtained previously was moved to the supplements (figure 3 —figure supplement 5). More data related to the trajectories using monocle and slingshot are displayed on figure 3 —figure supplement 4 and figure 3 —figure supplement 5 respectively.

'Additionally, the cephalopod-specific chromatophores, which are colored or iridescent, have been suggested to be modulated by the nervous system'. Maybe I am misunderstanding the sentence, but as I read it this is an understatement of what we know. Over 50 years of work have profiled the neural control of chromatophores in detail.

We reformulated this sentence to acknowledge the existing knowledge on the subject. It now reads: " Additionally, the cephalopod-specific chromatophores , which are colored or iridescent, have been shown to be modulated by the nervous system (Dubas et al. 1986; Wardill et al. 2012; Gonzalez-Bellido et al. 2014)" (lines 559-562).

Some of the supplementary figures are quite dark. Why not increase the brightness of the images? Others like Figure 4 supplement 1 are beautiful.

We have now increased the contrast on many images to help the reader better visualize the expression patterns shown in the figures.

Reviewer #3 (Recommendations for the authors):First of all, I would like to congratulate the authors on the extensive expression analysis study performed in this manuscript. However, I believe that this dataset offers so many more opportunities to better understand cell types in cephalopods, and in general, the data could be analysed and interpreted in more detail.Since the field of scRNAseq is moving incredibly fast, the authors should be careful with claims of being first (e.g. Abstract line 34). Preprints of scRNAseq data in the nervous system of other cephalopods have been recently published (Styfhals 2022, Gavriouchkina 2022, Songco-Casey 2022).

Thank you for mentioning this. We have now cited the recent single-cell RNAseq preprints that were missing from the manuscript (Lines 98, 656 and 679) and have removed the claims of being the first analysis of this kind.

The manuscript is missing key citations to support the results and to provide a comparative view of gene expression in other cephalopods. For example, the expression patterns of Elav, SoxB1, Asc have been reported during the development of octopus, squid, and cuttlefish, and could enhance the interpretation of the results observed in Loligo.

We added important citations related to the expression and function of these genes in other cephalopods as well as in other clades. We have developed the discussion to better confront the results obtained in our study of *Loligo vulgaris* to the existing literature (discussion: lines 682 – 733).

In general, the reader needs to be proficient in the brain areas of cephalopods in order to follow the results. Therefore, the manuscript would benefit from a short introduction of the main brain lobes in cephalopods to be accessible to a wider audience, either in the introduction or in the results in Figure 1C.

This is indeed a good point. An introduction to the brain areas that are mentioned in the manuscript is now available at the paragraph "molecular mapping of the *L. vulgaris* nervous system" (lines 253 – 266). To better understand the spatial arrangements of the lobes, we also provide a rough schematic of the anatomy in a new figure supplement (Figure 2 —figure supplement 1).

Please make sure to refer to the figures and figure panels chronologically. Multiple figure panels are not referred to throughout the text (e.g. the first figure described is Figure 1B and there is no notion to Figure 1A, similar to Figure 2, …).

Thank you for pointing this out. We have now checked the chronology of the figures, so that they appear in the right order in the text.

The authors distinguish 33 cell clusters when using 25 pca's and a resolution of 2. The UMAP, however, shows most cells in one single, central blob. This is striking knowing how different connective tissue is from neurons. Did the authors try other methods for clustering and dimensionality reduction, or change the parameters in the current algorithm to try to resolve the data better? It would also be helpful to specify the filtering parameters for filtering out low-quality cells in the methods section.

We have indeed noticed that the resolution of the clusters using different tissues (such as in the head) is quite different from resolving similar cells in the brain, which we have mainly focused on in the past. We have initially tested several values for PCs and resolution and felt 25 PCs and resolution 2 gave the best result. This issue of having clusters too close to each other could not be resolved using different clustering parameters (PCA selection method, number of PCs, cluster resolution). Representing the data with TSNE instead of UMAP also showed similar issues. We propose in the discussion that such lack of resolution, although it is sufficient to analyze cell types, is due to missing genes in the reference transcriptome (Results: lines 367-372, discussion: lines 652-666, 680-682). It is actually an intriguing point we frequently discuss which way might be more adequate: splitting or grouping clusters; this may be partly a semantic issue or a matter of taste and mostly decide on a middle path.

In the first section of the results, the authors list a handful of enriched genes used to differentiate the major cell type categories. However, little explanation is given to why these genes were chosen and a comprehensive overview supporting the claims is missing. It would help the reader if they would add a plot (e.g. a DotPlot) to support these data. In addition, a UMAP plot with cells colored according to cell cycle could clearly show where the progenitor cells are located on the UMAP.

This is indeed a good point. As mentioned above, the genes were chosen among a list of differentially expressed genes in the clusters. The choice among the top markers is made based on previous knowledge on the association of the gene to a specific cell type. For example, genes of unknown function that have never been studied in other species were not selected in our analysis because they could not inform us on the identity of cell types, even though they may be statistically relevant. The choice of markers was generally subjective based on our own interests and pre-existing knowledge as the systematic analysis of all the top markers would have been practically impossible in our case. To provide the reader with a better idea of the relevance of the genes chosen in this study, we now provide a dot plot showing the top markers across all cluster (Figure 1 —figure supplement 2) in addition to the full list of differentially expressed genes (supplementary table 1).

To avoid going back and forth between cell type categories in the first results paragraph, it would be helpful to describe Cl 27 (lines 202 onwards) and Cl 32 (lines 207 onwards) with the other nervous system cells, and Cl 29 (lines 213 onwards) with the epidermis section.

We now grouped the description of Cl27 and Cl32 with the description of neuronal markers in this section and the description of Cl29 directly following the description of the epidermis markers (results: lines 198-205).

In order to interpret HCR expression patterns, the authors should add axis labels to the figures. In addition, I understand that the figure panels are small to add annotation of the brain regions, but since we are looking at different levels through the brain, either a schematic with the figure or annotation would really help.

We agree that the pictures did not provide enough information for the reader to understand the exact location of the picture. In addition to the insets, we now also added axes labels and annotation of the anatomical structures visible on the micrographs.

The authors should explain the analysis performed to obtain the UMAP(?) plots in Figure 2C. Which cells were selected? Were they re-clustered? What do the clusters look like?

The UMAP plots on Figure 2C are obtained from a re-clustering of the cells highlighted in Figure 2A. We know added information about this in results and details about the parameters used to obtain this was now added in the materials and methods (lines 944957). The clusters obtained with this method are also shown with a UMAP on figure 3 —figure supplement 8.

Did the authors perform image manipulation e.g. Figure 2C and 2D for the DAPI channel? Why is DAPI absent from the neurons in the tetraspanin-8 HCR, or the serotonin transporter staining?

The images were not manipulated but this is the result of an experimental issue. All the HCR ISH experiments were performed on whole mount samples and sometimes we encountered problems with the penetration of DAPI inside the denser tissues. This is visible on several pictures where the outside of the animals is strongly labeled with DAPI but the signal fades as we go deeper in the tissue. We realize that this may confuse the reader and added a word of caution regarding this problem in the results and the concerned figure legend (lines 306-309, lines 1075-1077).

In Figure 3B, it would be helpful to add some genes involved in proliferation, as exemplified in the text.

We added new plots showing the expression of additional markers in the scRNAseq along the proposed differentiation axis. The genes in question are pcna, elf2-like, histone H3, ets4, hes-4, chymotrypsin, synaptogyrin, elav-like1, elav-like 2 and reflectin-8. These plots are available in Figure 3 -supplements 4 and 5.

The authors claim that neural stem cells undergo neuronal differentiation and then migrate to the nervous system. However, evidence is missing as the authors do not show where progenitor cells stop dividing and start to differentiate.

We agree that currently there is no data supporting this hypothesis. Although we show that cells in different stages are present in different locations, the timeline of differentiation and migration cannot be inferred from our data but we propose that such a stage exists and is yet to be characterized. We reformulated these statements to be more representative of the evidence. It reads as follows: " Together, the overlapping expression patterns of *e2f3, asc* and *soxB1* in domains associated with cephalopod neurogenesis strongly suggest that stem cells and neuronal progenitors are present the lateral lips of *L. vulgaris* suggesting that these cells are likely migrating to the nervous system at later differentiation stages " (results: lines 423-427 ).

The use of the marker foxD1 for differentiation neurons is not well supported. Additional marker analysis is required before this can be claimed.

This is indeed an important point, we apologize for not being clearer on this. In fact the, choice to include foxD1 in the analysis of neuronal differentiation was made after the HCR ISH experiments that showed its expression in the nervous system. In correlation with literature that link foxD1 genes with neuronal differentiation, we decided to check if we could computationally link the foxD^+^ clusters with neuronal clusters. We are aware that foxD1 genes are not a typical feature of differentiating neurons as it is also broadly expressed in other tissues in the development of other species (Herrera et al. 2004; Polevoy et al. 2017; Newman et al. 2018; Janssen et al. 2022). In the section about trajectory inference, we added additional markers of these cells (Figure 3 —figure supplements 4 and 5). We generally reformulated interpretations regarding foxD1 to avoid overstatements. While we are confident about this observation of where foxD1 is expressed in situ and in silico, we indeed agree that with the current data we may not define it as neuronal differentiation and have widely re-written this section and that it will in the future need to be thoroughly tested (discussion: lines 733-753).

The authors need to explain which clusters were used for trajectory analysis using Slingshot and how the data were reclustered. Were non-neuronal stem cells included? This might skew the analysis. Importantly, the method was not described in the methods section. It is critical for the readers to know which endpoints were selected in the slingshot algorithm. Off note, the slingshot is very dependent on clustering resolution and will try to connect all clusters. Have the authors tried a different pseudotime analysis method such as Monocle or FateID? Additionally, can the authors elaborate on which transcription factors are steering this differentiation? In the current state, the analysis is preliminary and contradicts present data on the origin and trajectory of neurons in the cephalopod brain (see comment below; Koenig et al. 2016, Deryckere et al. 2021).

Thank you for pointing out the lack of information regarding the trajectories and the need to replicate these results. We now include more detail about how the trajectories were calculated and which parameters were selected in the materials and methods in a new paragraph called "trajectory inference". We also include details about how the cells were chosen and re-clustered for figure 2 and 3 (Materials and methods: lines 944950). We now include in the main figure the analysis of the trajectories using monocle in addition to the previous analysis with slingshot that has been move to the supplements (figure 3 —figure supplement 5). The other cluster specific transcription factors that could be identified were also plotted onto the trajectory analysis on figure 3 —figure supplement 4 and 5. However, many genes that have been linked to neuronal differentiation (for example neuroD in Deryckere et al. 2021) could not be found in our data and are therefore not available to support our hypotheses. More generally, we agree that these results are preliminary and require further experimental validation (see discussion).

With the current interpretation of the data, the authors claim that foxd1 cells are an intermediate developmental state of neurons. This would be very striking since the cortex of the optic lobe has been described as being the inner retina. Additionally, it is contradicting the expression of amyloid β-binding like in the optic lobe cortex, which the authors say is only expressed in differentiated neurons. Is amyloid β-binding-like expressed in clusters 15, 19, and 20? How about other markers for differentiated neurons? What would all those intermediate cells do there? Do the authors think they migrate into the medulla or even the central brain? This is contradicting existing literature on the origin of neurons in the cephalopod brain (Koenig 2016, Deryckere 2021) and requires in-depth discussion.

We indeed agree on this point and as stated above we have included a more moderate interpretation. Our analysis of foxD1 originally showed two different things:

1. This gene is very specific to certain cell clusters and is highly expressed in these cells compared to other clusters (Figure 3A). These cells cannot be directly assigned a cell-type identity as they lack known cell-types markers, including markers of differentiated neurons.

2. This gene is highly expressed in different regions of the nervous system, including optic lobe cortex (Figure 3, figure 3 —figure supplement 2). Importantly, we do not think that the expression of this gene in these regions necessarily means that they are differentiated neurons.

Because these two observations do not clearly indicate a link between foxD1 and neurons, we decided to computationally assess whether there might be a developmental relationship between the cell types found in the nervous system and the progenitor-like cells present in the lateral lips. Contrary to our expectation, the trajectory analysis put foxD1 cells in between putative progenitors (expressing e2f3, asc and soxB1) and differentiated neurons (expressing elav-like 1, elav-like 2, tetraspanin-8, amyloid-β-binding) (We do find some cells expressing low levels of amyloid β-binding-like in FoxD expressing cells (clusters 16, 19, and 30)). Essentially, it means that there are more transcriptional similarities between foxD1 and neurons (respectively between foxD1 cells and asc^+^ cells) than there are between asc^+^ cells and neurons. Whether that means that foxD1 expression marks a specific stage neuronal differentiation is not clear and will needs to be further investigated. We moderated our conclusions regarding the analysis of foxD^+^ in the text and provide more discussion of the subject (lines 733-753). We agree that a theoretical intermediate cell stage could in any case not be defined by the expression of a single gene and that thorough analysis of more transcription factors would be necessary to support these claims. The absence of many neurogenesis-associated transcription factors from our data renders this analysis more difficult and a higher resolution dataset may be able to conciliate our findings with previous literature.

The authors prepared beautiful summary diagrams in Figures 3 and 4. However, they should be referred to in the main text.

Thank you for pointing this out, we have now made sure to refer to the diagrams in the text (lines 460, 553)

After performing extensive HCR expression studies, can the authors go back to their scRNAseq data and better annotate the different clusters?In order to better understand neuronal diversity and neural cell types (as the authors claim), the dataset has to be analyzed in more depth. 33 clusters/cell types seem low given the number of cells that were sequenced and the different tissues that were sampled. Can the authors identify the different brain regions when subclustering the neurons? Can they identify and annotate different neural cell types?

This is an important point and we indeed hoped to find genes identifying neural subtypes or defining certain brain regions. We indeed identified some genes with regionalized expression, such as scratch and LIM, which to some degree can be associated to certain clusters (scratch 5, 15, 20; Lim high in 22). However, we did not uncover wide diversity of neuron subtypes. This may be partly due to the fact that we took entire heads for our analysis (as compared to only brain) and partly to the coverage of the transcriptome.

The HCR ISH studies provided a lot of additional information about the genes originally described in the scRNAseq data and the cell types that they represent. The interpretation of the results combining these two methods provides avenues for future studies but is usually not sufficient to confirm the identity of a previously undescribed cell type. The clustering parameters used initially resulting in a total of 34 clusters could have been changed to obtain more clusters but the markers genes of these new clusters could not have informed us on the possible identity of these cell types. The example of the lack of resolution for the nervous system well exemplifies this limitation in our study: several genes were shown to be expressed in specific lobes of the nervous system but this did not correlate with specific clusters in the scRNAseq data. We envision in the future to extend our investigations by including more developmental stages (and behavioral context) and to possibly assembling a *Loligo* genome to map the same reads to may resolve this issue (Results: lines 369-372, discussion: lines 652-666, 680-682).

The authors present several striking findings. However, at the moment, the Discussion section explains the results obtained in the paper, but does not place the acquired data within the current state-of-the-art. Parts of the results should be moved to the discussion and the discussion should be elaborated on.

We rewrote the discussion to better contextualize our findings within the existing knowledge about cephalopod cell types and the different functions of the genes discussed in this paper. We also cited many additional papers to better support our interpretation of the data.

Specific comments:Line 76: "by the use its specialized iridescent cells" misses an ofLine 97: it would be helpful for the readers to add the "pre-hatchling stage" to stage 28, as was done in the Results section for the audience to have a better idea of what stage was studied.

We now made sure to specify pre-hatchling in every mention of the specific stages (lines 102, 133, 432, 835, 987).

Throughout the text (starting line 219) please write "HCR ISH" instead of just "HCR". The chain reaction method is not specific to in situ hybridization and can also be used in immunohistochemistry and others.Line 101: "in all animals" please be more specific; not all animals have neurons.

We corrected this sentence to "in most bilaterians".

Following the recommendation of the reviewer, we now refer to the experiments as HCR ISH experiments.

Line 110: "marking distinct stages of neural development", neural differentiation would be more appropriate.

We corrected this and changed to "neural differentiation".

Line 126: Add a reference to Figure 1A to show what the head region is.

The dotted line on figure 1A represents the plane of amputation. Everything anterior to that was used for these experiments. We made that clearer in the legend of figure 1A: " The dotted line represents the plane of amputation. The region anterior to this line is what was used in this study and is referred to as the head region."

Line 133 says 34 cell clusters were identified, but Figure 1B shows 33.

We kept the default counting system used by Seurat where the count starts at 0. That is why there are 34 clusters but they are numbered to 33. This is specified in the text as follows: " This resulted in a total of 34 cell clusters numbered from 0 to 33" (lines148149).

Line 145: "smooth muscle-like cells (Cl26)" should be Cl25.

We thank the reviewer for noticing this mistake, the number was changed to 25 (now on line 159).

Line 156-159: add a reference for this statement, maybe a specific example.

We added the following citations to the statement referring to previous studies of connective tissue in cephalopods: Thompson and Kier 2001, Kier and Stella 2007.

Line 167-171: "a gene whose function is homologous to …" and " exact function … is not known". This is contradictory. Do you mean that the gene's sequence is homologous?

This sentence was reformulated to explain that the function of the *Drosophila melanogaster* papilin ortholog has been studied but the function in other species is unknown. It reads as follows: "… *papilin,* a gene which was suggested to play a role in the formation of extracellular matrix in *Drosophila melanogaster* (Fessler et al. 2004; Apte 2020). The function of this gene in other clades is however not known" (lines 185188).

Line 199-201: with the comments below on SoxB1 and Ascl1, this sentence needs to be revisited.

This sentence was revisited according to the discussed roles and expression patterns of soxB1 and asc: " The presence of a *soxB1* in these cells indicates that they may proliferating neuronal progenitors, based on previous evidence of expression of soxB1 genes during neurogenesis in cephalopods (Miyagi et al. 2009; Focareta and Cole 2016)" (line 227-230).

Line 239-240 and also further in the text: "in the medial portion of the optic lobes" do the authors mean the medulla (in contrast to the cortex (inner and outer granular layer))? Medial portion is not clearly defined and the cortex also stretches quite medially. (also line 278, 540).

We did indeed mean the medulla when mentioning the medial portion of the optic lobes. We now specifically refer to the medulla and cortex in the text to use the correct terminology and avoid unnecessary confusion.

Lines 263 and 271: please rearrange the supplementary figure so you can refer to panels A-D first. In general, please refer to figures chronologically.

The figure was rearranged to match the chronology of the text.

Line 298 with Figure 2D: please switch panels of LIM and serotonin, so they follow the text.

The panels were switched in order to match the text (figure 4).

Line 326: this part covers more than stem cells and progenitors (line 375 and onwards).

This section was renamed stem cells and neuronal differentiation to include the proposed cell types of the neuronal lineage.

Line 327: please add an introductory sentence to this new paragraph; rephrase the first sentence (grammatically incorrect).

We reformulated this sentence to " Several marker genes expressed in specific clusters suggest an involvement in proliferation and possess some conserved markers of neuronal differentiation". We also added an introduction to the paragraph (line 376382).

Line 338-340: this is an overstatement. What identity are the authors referring to? Genes involved in proliferation etc do not give identity to stem cells and Cl02 and Cl14 are also actively cycling. Do the authors suggest only Cl10 and Cl17 are stem cells? Can the authors show more evidence?

This sentence was modified and references were added to support the claim. We indicate that cells of Cl02, Cl10, Cl14 and Cl17 are all proliferating cells. E2f3 is highly expressed in Cl10 and Cl17 and is also co-localized with asc in the lateral lips. The sentences were revisited as follows: " The role of *e2f3* to promote cellular proliferation has been shown to be conserved between *D. melanogaster*, *C. elegans* and mammalians (Dynlacht et al. 1994; Attwooll et al. 2004; DeGregori and Johnson 2006). Because these clusters highly express proliferation markers but lack any distinctive features of differentiated cell types, we propose that they may be stem cells" (lines 392-397).

Lines 350-359: I would suggest moving this part to the discussion and incorporating the following comments:Lines 347-352: The expression of Asc is not as well conserved as the authors suggest. Indeed, it has some proneuronal role, but the level of differentiation in the cell it is expressed is very different across the animal tree. In *Drosophila* as in other non-insect arthropods, Asc is expressed in quiescent ectodermal cells to drive them into the neural lineage (formation of a neural progenitor cell) (e.g. Cabrera et al., 1987; Skeath and Carroll, 1992). In contrast, in vertebrates and also in spiralia such as capitella teleta, Ascl1 is expressed in neural progenitor cells to drive them into differentiation (e.g. Bertrand 2002; Sur 2020). Ascl1 expression has been studied in octopus and seems to have a similar function as in capitella, not *Drosophila* (Deryckere 2021). This scRNAseq dataset has the power to test this hypothesis and figure out whether, in cephalopods, as is expressed in proliferating and not quiescent cells.Lines 354-355: this conclusion does not reflect the data shown. The authors are showing proliferating stem cells, not the location of differentiation. In octopus, differentiation seems to happen in the transition zones (NeuroD cells, Deryckere 2021). It would be interesting to locate those cells in your dataset.Lines 356-359: Again here, one needs to be careful to extrapolate the functionality of a gene in one species to other species. In invertebrates, SoxB1 is also expressed in differentiated neurons (e.g. Le Gouar et al., 2004; Semmler et al., 2010). Additionally, expression of SoxB1 has been described in Sepia (Focareta and Cole, 2016) and octopus (Deryckere 2021), indicating expression in neurons (with some interesting discrepancies). Furthermore, be careful with the use of the word "neuroblast" because it refers to different types of cells in different animals.Line 361: "Interestingly" it seems a general theme for cephalopods, and maybe invertebrates in general (see comment above).Line 367: Data are missing to show where the neural progenitor cells differentiate.Line 370-374: in line with this, SoxB1 expression is also found in sensory epithelia of sepia and octopus (e.g. suckers and statocyst; Baratte, S. and Bonnaud 2009; Deryckere 2021).

We thank the reviewer for these very helpful comments. We know discuss the discrepancies in the role of asc across different phyla and can replace our findings in this context (lines 709-721):

"We interestingly observed the expression of *asc* in the same region, together with several proliferation markers. Extensive study of neurogenesis in *Drosophila melanogaster* has shown that the proneural complex *asc* can determine the neuronal fate of quiescent multipotent ectodermal cells (Campuzano and Modolell 1992; Skeath and Doe 1996). In vertebrates on the other hand, *asc* is expressed in cells that have already undergone neuronal specification but are self-renewing (Bertrand et al. 2002; Soares et al. 2022). Studies of neurogenesis in the annelid *Capitella teleta* has shown that, similarly to vetrebrates, proneural genes such as *asc* are expressed in self-renewing neuronal progenitors (Sur et al. 2020). Our data suggests that *asc* is also expressed in proliferating cells, similiarly to vertrebrates and *Capitella teleta* but unlike *Drosophila*.”

We also discuss the role of soxB1 and compare it to what has been observed in other cephalopods but also refer to the documented expression of this gene in differentiated neurons (lines 721-733):

“The observation of *soxB1* expression together with *asc* strengthened their characterization as neuronal progenitors due to the documented role of this gene in neuronal differentiation (Miyagi et al. 2009). However, strong soxB1 expression in differentiated neurons has been shown in the gastropod *Patella vulgate* and in the cephalopods *Sepia officinalis* and *Octopus vulgaris* (Le Gouar et al. 2004; Focareta and Cole 2016; Deryckere et al. 2021). In *L. vulgaris* we observe similar results where *soxB1* is expressed in the lateral lips together with *asc* but is also expressed in differentiated neurons. This was consistently shown by the HCR ISH experiments as well as the single-cell analysis. Therefore, the hypothesis proposed for the action of *soxB1* in the early stages as well as the later stages of neuronal differentiation in *Octopus vulgaris* is consistent with our data (Deryckere et al. 2021).”

We also discuss the possible limitation of our analysis as some genes that have been shown to be involved in *Octopus vulgaris* neurogenesis could not be found in our data (lines 742-753):

“Such a role has been proposed in Octopus vulgaris in the cells expressing *neuroD* that are expressed in a gradient between the lateral lip and the central nervous system, which may indicate a pattern of migration of differentiating neurons (Deryckere et al. 2021). We could however not correlate these results with our hypothesis of an intermediate *foxD1*-expressing stage because *neuroD* expression could not be detected in specific clusters in our single-cell dataset. Although the trajectories we calculated point towards a role of FoxD1 in neuronal differentiation, the relatively poor cluster resolution and the lack of cells along the trajectory axes shown in figure 3 suggest that this hypothesis would require further investigation that should include a more complete set of genes that have been involved with cephalopod neurogenesis.”

A short interpretation of the data is left in the results to facilitate comprehension of the data but is then discussed more in depth in the discussion.

Line 376-378: add a reference and elaborate. Regulators of development is a very general process. In addition "for this reason, these cells were presumed to be differentiating cells": this is an overstatement and needs additional gene expression patterns to support this presumption.

We corrected this sentence to not overstate the identity of these cells. Although there is evidence of foxD1 being involved in the development of the nervous system, the hypothesis of these cells being a possible stage of neuronal differentiation is only prompted by the expression pattern observed by HCR ISH followed by the trajectory analysis described later in the manuscript. The sentence was modified as follows: " FoxD genes have been documented to be expressed in the nervous systems of *Mus musculus*, *Xenopus laevi*, *C.elegans* and *D. melanogaster* and to play a role in the development of neuronal cell types although not exclusively (Herrera et al. 2004; Polevoy et al. 2017; Newman et al. 2018; Janssen et al. 2022)" (lines 437-441).

Line 382-385: optic lobe cortex = inner and outer granular layers, please adjust and rephrase. From Figure 3C upper panel, it seems that Foxg1 is also expressed in the medulla. Or is this all cortex? Annotation would be helpful here.

This sentence was revisited to describe the expression pattern that is indeed observed strongly in the cortex but also in the medulla. The sentence reads al follows: " HCR ISH assessment of the expression of *foxD1* showed that these cells are strongly expressed in the cortex of the optic lobes and in the medial basal and anterior basal lobes (figure 3C). The expression of *foxD1* is very strong in the cortex of the optic lobes and in the medial and anterior basal lobes but is also sparsely located in the medulla (figure 3 —figure supplement 2)" (lines 441- 446).

Line 386: this is an overstatement. Most neurons in the brain express elav1 and the function of foxD1 is very broad. Additional markers (like for example NeuroD) are required to support this claim.

This sentence was revisited as follows: " The expression pattern of *foxD1* indicated that these cell types are present in the nervous system and could therefore be types of neurons or neuronal progenitors" (lines 446-448). The claims about the role of foxD1 are stated later in the manuscript and are discussed in a critical manner (lines 733753).

Lines 394-400: Figure 3D is not mentioned in the text and seems substantial. The authors might also want to revise it with the comments above and in public review.

We modified figure 3D to better represent that soxB1 is expressed in both the lateral lips and in the nervous system together with markers of differentiated neurons. The HCR ISH experiments do not by themselves support the presence of an "in-between" stage as was suggested by the figure previously.

Lines 408-411: please add references to support these statements.

We have added references that support the conservation of the phototransduction cascade between cephalopods and other protostomes. The sentence reads as follows: " The phototransduction cascade that allows light stimuli to be transformed into an electrical signal in the neurons is well conserved among cephalopods and we were able to clearly identify photoreceptors in the scRNAseq data (Davies et al. 1996; Monk et al. 1996; Arendt 2003)" (lines 510-514).

Lines 419-421: Why did the authors choose to map the expression of Pax9? Koenig et al. studied the eye in Doryteuthis and mapped the expression of several genes important for eye development. A reference to this work would be appropriate here. Pax9 is not an eye gene in vertebrates and invertebrates (in contrast to Pax2/6). In which clusters is it expressed (again here, a dotplot of the genes under study would be helpful)? The level of the image in Figure 4A for Pax9 is also different than the others, why is this? In which cells of the eye is Pax9 expressed?

Pax9 was remaining for an initial screen of transcription factors that may be related to developmental processes. After careful re-analysis we decided to exclude this gene from the analysis as it is not a cluster-specific marker in the scRNAseq data and the expression pattern revealed by HCR ISH showed expression in what appeared to be non-neuronal tissue around the eye. Because there was no pre-existing evidence of this gene being involved in sensory cells, we realized that including of this gene in our analysis was not justifiable.

Line 448: Figure 5B should be Figure 4B I think.

This is indeed the case. We changed it to 4B.

Line 457: can the authors verify that they selected cells with expression level >3? From the legend in Figure 4D, the expression level of 3 seems present in a lot of cells (it looks more like the selected level was 5). In addition, the authors should explain how they obtained the graphs on the right. It looks like they underwent an extra transformation? Or are they just distorted?Line 459, and together with the comment above: It would be helpful to be able to check the cluster numbers. For example, are cilia-associated protein cells from Cl27? A dotplot or heatmap might be more convincing to support these statements.In addition, please add a reference to Figure 4D here.

It was pointed out to us in other comments out that the logic in validating the sensory cells this way was logically flawed. We have replaced the graph on figure 4D by plots of the expression of the sensory genes within the neuronal subset previously used in figure 2 (figure 2 —figure supplement 8, materials and methods – lines 944-948). This way, we show that the sensory cells are all included in the neuronal clusters although not all of them are cluster specific (for example *fmrf,* figure 2, figure 2 —figure supplement 8).

Line 468-470: egf-like does not seem to be expressed in Cl03 and 31 from Figure 5A as suggested in the text.

We accidentally mixed up some of the clusters and genes in this section. We thank the reviewer for spotting these mistakes. egf-like if strongly expressed in cl08 and cl09 whereas Cl03 and Cl04 are characterized by the expression of papilin. Cl31 expresses the epithelial splicing regulatory protein esrp.

Lines 471-493: please add the discussed genes to the dotplot in Figure 5A or supplement.

We now added the additional genes discussed in the text to the dotplot in figure 5A

Line 541: It seems that cells in Cl27 do express neuronal markers. Please revise.

We revised the sentence and stated the expression of neuronal markers in Cl27 as described in other sections. The sentence reads as follows: " While *madnf-*expressing cells of Cl27 co-express neuronal markers, the cells of Cl23 and Cl31 do not" (lines 619-620).

Line 545 and onwards: please not only discuss the results obtained but place the discussion in the state-of-the-art literature. Several statements have already been proposed in other papers which deserve to be referenced (e.g. lines 559, 568). Multiple genes mapped with HCR in this manuscript have been described in other cephalopods and assigned a preliminary function (e.g. Buresi 2013, 2014; Shigeno 2015; Koenig 2016; Focareta 2016; Deryckere 2021 and others). Please acknowledge their presence in appropriate sections, and, if the authors which, they might add a paragraph describing consistency or differences in expression.

As mentioned in response to public reviews, we have extensively developed the discussion to include more references, among which we can place our results.

Line 562: in line with previous comments in the results, please revise. This is based on the functionality of asc in *Drosophila*, but not other animals. Additionally, in the opinion of the authors, what are the asc+ e2f3- cells?

We now discuss these results, taking into account the differences in the function of asc across different animal phyla (lines 709-721). We propose that these cells are early self-replicating neuronal progenitors.

Line 713: please add the parameters used to filter low-quality cells.

The parameters used are now included in this section (lines 928-932). In addition, plots relevant to the assessment of the quality of the scRNAseq data are now provided in figure 1—figure supplement 1.

Line 722: please add additional information on how cells were subclustered (and re-clustered?). Please add a description of the slingshot analysis.

We now provide more detailed information about the clustering parameters for the full dataset as well as the subclustering used in figure 2,3 and 4 (lines 944-957).

Figures:In general, please rearrange so each panel comes chronologically in the Results section. Make sure to mention each panel in the text.Add axes to the confocal images and consider annotation.

We have now provided annotation and axes to facilitate the reader's understanding of the data. The panels have been rearranged to match the text.

Figure 1:A) Is this the ventral view instead of the dorsal view? The funnel is visible?

This is indeed a ventral view, we now fixed this mistake.

Please arrange the abbreviations alphabetically. Ibl and sbl are missing.Figure 2:Line 790: UMAP with clusters.C) Add legend to the 1→6 expression? Bar. It is not clear how these featureplots were calculated. Which clusters were selected? Were they re-calculated? Please show the dimplot with cluster annotation. What is on the x- and y-axes? UMAP? t-SNE?D) Please rephrase 'that were not represented in the scRNAseq data': do the authors mean not differentially expressed? Were the genes absent from the reference transcriptome?Figure 3:A) Why are Cl2 and Cl14 encircled?C) Explain what the red encircled areas are.

We have added the missing information in the figure legends.

Figure 4:D) Double check that 3 is the threshold. The curly bracket is confusing (it reads like the right panels feed into the left). What are the x- and y-axes? Similar comment to Figure 2C.

As mentioned above figure 4D was modified to show the expression of the genes in the subset of neurons previously used in figure 2.